# Estimating statistical errors in retrievals of ice velocity and deformation parameters from satellite images and buoy arrays

Wolfgang Dierking[1,2], Harry L. Stern[3], Jennifer K. Hutchings[4]

[1]Center for Integrated Remote Sensing and Forecasting for Arctic Operations, University in Tromsø, 9019 Tromsø, Norway

[2]Alfred Wegener Institute Helmholtz Center for Polar and Marine Research, 27570 Bremerhaven, Germany

[3]Polar Science Center, Applied Physics Laboratory, University of Washington, 1013 NE 40[th] Street, Seattle, WA 98105, USA

[4] College of Earth, Ocean, and Atmospheric Sciences, Oregon State University, 104 CEOAS Administration building, Corvallis, OR 97331, USA

*Correspondence to*: Wolfgang Dierking (Wolfgang.Dierking@awi.de)

**Abstract.** The objective of this note is to provide the background and basic tools to estimate the statistical error of deformation parameters that are calculated from displacement fields retrieved from synthetic aperture radar (SAR) imagery or from location changes of position sensors in an array. We focus here specifically on sea ice drift and deformation. In the most general case, the uncertainties of divergence/convergence, shear, vorticity, and total deformation are dependent on errors in coordinate measurements, the size of the area and the time interval over which these parameters are determined, and the velocity gradients within the boundary of the area. If displacements are calculated from sequences of SAR images, a tracking error also has to be considered. Timing errors in position readings are usually very small and can be neglected. We give examples for magnitudes of position and timing errors typical for buoys and SAR sensors, in the latter case supplemented by magnitudes of the tracking error, and apply the derived equations on geometric shapes frequently used for deriving deformation from SAR images and buoy arrays. Our case studies show that the size of the area and the time interval for calculating deformation parameters have to be chosen within certain limits to make sure that the uncertainties are smaller than the magnitude of deformation parameters.

## 1 Introduction

Sea ice drifts under the influence of wind and ocean currents. Spatial gradients in the sea-ice motion lead to distortion of the sea-ice cover, termed deformation. The retrieval of sea ice drift vectors and deformation parameters from pairs or sequences of satellite synthetic aperture radar (SAR) images has gained increased attention during recent years because of the growing availability of suitable data (e.g. Stern and Moritz, 2002; Karvonen, 2012; Berg and Eriksson, 2014; Komarov and Barber, 2014; Lehtiranta, 2015; Muckenhuber et al., 2016; Demchev et al., 2017; Korosov and Rampal, 2017). Sea ice kinematics is also studied based on data from arrays of buoys or GPS receivers (e.g. Lindsay, 2002; Hutchings et al., 2008; Hutchings et al., 2012; Itkin et al. 2017), which in addition can serve as reference in comparisons to motion vectors obtained from SAR images. The knowledge of spatially detailed motion and deformation fields is potentially useful in ice navigation to locate divergent or compressive ice areas, as complementary information for operational sea-ice mapping, for validation of models for forecasting of ice conditions, and for assimilation into ice models (Karvonen, 2012). Such practical applications require that the errors of the retrieved drift and deformation parameters are known. For buoys, errors in drift measurements depend on the accuracy of position and time readings. The accuracy of deformation parameters is not only affected by errors in drift magnitude and direction but also by the size and shape of buoy arrays (e.g. Hutchings et al., 2012; Griebel and Dierking, 2018). Drift vectors derived from pairs of satellite images are the result of correlation techniques or object detections, while deformation parameters are calculated from spatial arrangements of adjacent drift vectors surrounding the area of interest, in a manner that is independent of the coordinate system. This means that drift and deformation errors do not only depend on the geolocation accuracy and spatial resolution of satellite images but also on the reliability and robustness of the drift retrieval

algorithm. In this technical note we focus on the estimation of statistical errors for ice velocity and deformation. The issue of error estimation was repeatedly addressed in the past, scattered in a number of publications and restricted to single aspects related to the respective analysis (e.g. Lindsay and Stern, 2003; Hollands and Dierking, 2011; Bouillon and Rampal, 2015; Hollands et al., 2015; Linow et al., 2015; Griebel and Dierking, 2018), and is also addressed in a more recent analysis by Bouchat and Tremblay (2020). Our motivation is to provide the mathematical background, together with examples of applications and discussions of validity, in a broader context. We emphasize that here we deal with statistical errors, but not with boundary definition errors as described, e.g. in Lindsay and Stern (2003), Bouillon and Rampal (2015) and Griebel and Dierking (2018). Although this note is specifically focused on retrievals of parameters characterizing sea ice kinematics, the mathematical framework is also applicable to movement and deformation of ice shelves and glaciers, or for model simulations of sea ice, glacier, and ice sheet dynamics.

In Sect. 2 we summarize the basics and provide equations for calculating errors of drift and deformation parameters: divergence, vorticity, shear, and total deformation. The equations are used in Sect. 3 to quantify the influence of different parameters such as geolocation and tracking errors, or shape and size of buoy arrays and grid cells. Conclusions are presented in Sect. 4.

## 2 Errors of drift and deformation parameters

In this section, we provide a short description of the estimation of errors and the computation of strain rates, and then derive the statistical errors for drift velocity, polygon areas, divergence, shear, vorticity, and total deformation. The statistical errors quantify uncertainties that are introduced by random fluctuations in the measurements. If the random fluctuations are small, data are measured with a high degree of precision, but not necessarily with high accuracy. The latter requires that the measured value is close to the true value, whereas precision refers to the reproducibility of a measurement (Bevington and Robinson, 2003, chapter 1).

### 2.1 Error propagation and calculation of deformation

The formula for error propagation is based on the splitting method, i.e. the decomposition of a measured variable $x$ into its true value and the measurement error: $x = x_{true} + x_{error}$, where $x_{true}$ is considered to be a constant, and $x_{error}$ is a random variable with expected value $E(x_{error}) = 0$ and variance $E(x^2_{error}) = \sigma^2$. If a quantity $Q$ is calculated from measured variables $x_k$, i. e. $Q = f(x_1, x_2, ..., x_n)$, a Taylor series expansion can be applied to estimate the error of $Q$. Usually only the linear term is retained:

$$Q = f(x_{1,true}, x_{2,true}, ..., x_{n,true}) + \sum_{i=1}^{n} \left[ \frac{\partial f}{\partial x_i} (x_{1,true}, x_{2,true}, ..., x_{n,true}) \right] [x_{i,error}] \tag{1}$$

The variance is obtained by moving the first term to the left-hand side, squaring both sides and applying the expected value operator E( ) (Bevington and Robinson, 2003). This operation results in

$$\sigma_Q^2 = \sum_i \left( \frac{\partial f}{\partial x_i} \right)^2 \sigma_i^2 + \sum_{i \neq j} \sum \left( \frac{\partial f}{\partial x_i} \right) \left( \frac{\partial f}{\partial x_j} \right) \sigma_{ij} \qquad i=1,n, j=1,n \tag{2}$$

where $\sigma_i^2$ is the variance of $x_i$ and $\sigma_{ij}$ the covariance of $x_i$ and $x_j$. If we can assume that the errors are uncorrelated, the second term on the right side of (2) is zero. We will use the notation "uncertainty" synonymously with "standard deviation of the absolute error".

Deformation parameters are calculated from different combinations of the components of the velocity gradient tensor ($\partial u/\partial x$, $\partial v/\partial x$, $\partial u/\partial y$, $\partial v/\partial y$) = ($u_x$, $v_x$, $u_y$, $v_y$) (Leppäranta, 2011), here given in a Cartesian coordinate system, where $u(x,y)$ and $v(x,y)$ are the velocity components in $x$- and $y$-direction at position $(x,y)$. We have

divergence $\qquad\qquad \dot{\varepsilon}_{div} = u_x + v_y$ (3a)

vorticity $\qquad\qquad \dot{\varepsilon}_{vrt} = v_x - u_y$ (3b)

shear $\qquad\qquad\qquad \dot{\varepsilon}_{shr} = \sqrt[2]{\left(u_y + v_x\right)^2 + \left(u_x - v_y\right)^2}$ (3c)

and total deformation $\qquad \dot{\varepsilon}_{tot} = \sqrt[2]{\dot{\varepsilon}_{div}^{\,2} + \dot{\varepsilon}_{shr}^{\,2}}$ (3d)

Divergence and shear are the two invariants of the symmetric deformation tensor. The dimension of $\dot{\varepsilon}$ is velocity change per length unit, hence [time]$^{-1}$. For ease of reference, we briefly repeat the physical meaning of different velocity gradient combinations (after Cuffey and Paterson, 2010; Leppäranta, 2011): Imagine a rectangle with its sides $L_x$ and $L_y$ parallel to the

$x$ and $y$-axes of a 2D Cartesian coordinate system. In this case the gradients $u_x$, $v_y$ are normal strain rates, leading to an extension or contraction of the rectangle in the respective direction. The normal strain along the $x$-axis, e.g., is $\Delta L_x(t) / L_x = u_x \Delta T$. Here $\Delta T$ is the time interval $\Delta T = t - t_0$ during which the effect of deformation is analyzed, and $L_x + \Delta L_x$ is the side length at time $t_0 + \Delta T$. The sum $u_x + v_y$ is the divergence or convergence, dependent on the sign. The expression $u_y + v_x$ is linked to the change of shape of the rectangle (pure shear). The normal shear, $u_x$ - $v_y$, quantifies the change in length difference between the sides of

the rectangle. The vorticity ($v_x$ - $u_y$), which is twice the rotation rate, describes the rotation about an axis vertical to the $x$-$y$ plane (positive counterclockwise) without change of shape. Let the rectangle be located in a temporally constant velocity field with, e.g., $u_x = 0.1$d$^{-1}$, $v_y = 0.05$d$^{-1}$, $u_y = 0$, $v_x = 0$, then the divergence is $\dot{\varepsilon}_{div} = 0.15$ d$^{-1}$ = 15% d$^{-1}$. Assuming that the sides of the rectangle are $L_x$ and $L_y$ at time $t_0$, its area $A_0 = L_x L_y$ increases to $(L_x + u_x L_x \Delta T) (L_y + v_y L_y \Delta T) = A_0 (1 + u_x \Delta T)(1 + v_y \Delta T) = 1.155 A_0$ for $\Delta T = 1$ day. Since only the difference $u_x$-$v_y$ contributes to the square root (3c), $\dot{\varepsilon}_{shr} = 0.05$d$^{-1}$=5% d$^{-1}$ is the normal shear

(Hutchings et al., 2012).

The deformation of a region $R$ (covered by the buoy array or grid cell) with area $A$ is calculated from the *spatial averages* of the velocity gradient components over the region $R$, in Eq. (4) indicated by an overbar. For the $u_x$ component, for example, the expression is (Thorndike, 1986):

$\overline{u_x} = \frac{1}{A} \iint_R \frac{\partial u}{\partial x} da = \frac{1}{A} \oint_C u\, \boldsymbol{n}\, \boldsymbol{e_x}\, dl$ (4)

Here $da$ and $dl$ are the differentials for area and length, $\boldsymbol{n}$ is the outward normal vector to the perimeter $C$ of $R$, and $\boldsymbol{e_x}$ is the unit vector in $x$-direction. This is Green's theorem, which relates a line integral along a closed curve $C$ to the area integral over a plane region $R$ bounded by $C$. The application of the theorem requires that the velocity components $u$ and $v$ have continuous

first-order partial derivatives on $R$. In a Cartesian coordinate system, the calculation of the velocity gradient in x-direction is carried out using

$u_x = \frac{1}{A} \oint_C u\, dy \cong \frac{1}{2A} \sum_{i=1}^{n}(u_{i+1} + u_i)(y_{i+1} - y_i)$ (5)

and the other components of the velocity gradient tensor accordingly. In Eq. (5) we have omitted the overbar above $u_x$. The sum comes from the trapezoid rule for integration, taking $n$ points around the perimeter of $R$, where $(u_{i+1} + u_i)/2$ is the estimate of $u$ on the $i^{th}$ segment, $(y_{i+1} - y_i)$ is $dy$, $i$ is the summation index which traces the boundary in a counterclockwise sense, $n$ is the number of vertices for the grid cell (or number of buoys), and $A$ is the area of the grid cell (or of the polygon spanned by the buoy array). Here, $u_{n+1} \equiv u_1$ and $y_{n+1} \equiv y_1$ (closed polygon). The velocity gradients are implicitly averages over $R$. This will

also be the case for our estimates of the deformation parameters, Eqs. (3a) – (3d).

The velocity vectors may be obtained from an array of buoys, where the buoys' positions are regarded as the vertices of a polygon. The displacement of a buoy is usually calculated from the distance between distinct positions, and the velocity is determined as the displacement divided by the time period between position fixes. When using pairs of satellite images, sea ice deformation is obtained from the displacements of recognizable structures or patterns in these images. These are referred to as ice structures from here on. In the reference image, a grid can be constructed by connecting the center positions of adjacent ice structures by lines. If movements of single ice structures differ between acquisitions of image 1 and image 2, the shapes and sizes of grid cells have changed in the second image. It is the presence of velocity gradients due to locally varying physical forces that causes the deformation. In practice the movement of sea ice is obtained using different methods (e.g. Holt et al., 1992; Stern and Moritz, 2002; Karvonen, 2012; Muckenhuber et al., 2016; Korosov and Rampal, 2017), which determine the spatial distribution and density of the displacement vectors. The vectors can be regularly spaced on the crossing points of horizontal and vertical grid lines as a result of pattern matching algorithms in an Eulerian approach, or they can be irregularly distributed, which is typical for the Lagrangian approach applied in feature or buoy tracking (see Fig. 1).

The errors discussed in the following subsections can be traced back to errors in the position of reference points (i.e. vertices of a grid, or buoys). Lindsay and Stern (2003) denote this error type as geolocation error. On a horizontal plane two coordinates (e.g $x$, $y$ or latitude, longitude) determine the positions of the start and end points of the displacement, respectively. The distance $d = \sqrt[2]{(x'-x)^2 + (y'-y)^2}$ is prone to the errors of the coordinate readings. Its uncertainty is $\sigma_d^2 = 2\sigma_{coord}^2$, assuming $\sigma_{coord} = \sigma_x = \sigma_y = \sigma_{x'} = \sigma_{y'}$, and no correlation between coordinate measurements at the end points (see Eq. (2)). When displacements are retrieved from a pair of SAR images, one needs to consider position and tracking uncertainties, i.e. $\sigma_{coord}^2$ and $\sigma_{tr}^2$, respectively. The latter arises from the fact that in a satellite image details of structures on pixel scale may be difficult to match between images 1 and 2. In this case the uncertainty in displacement (which here is the distance between positions of a fixed point on an ice structure in images 1 and 2) is $\sigma_d^2 = 2\sigma_{coord}^2 + \sigma_{tr}^2$. For buoy arrays, $\sigma_{tr}^2$ is zero, since a buoy remains fixed relative to the ice floe on which it was deployed.

In a SAR image, the geolocation (position) error is caused by the inaccuracies of the parameters describing the satellite orbit as a function of space and time. In general, the error caused by these inaccuracies is uniform across the image with only small local variations. Hence the assumption of independent geolocation errors is not valid if distances between moving objects are small. Holt et al. (1992) give a correlation length of 10 km for the uncertainty of the geolocation error, $\sigma_{coord}$, but correlation lengths of up to 100 km may be possible (R. Kwok, personal communication, 2020). Deformation parameters from SAR image pairs are usually calculated over regions that are on the order of 10 kilometer or less across. With correlation lengths of ≥10 km, geolocation errors at all pixels in the region are almost equal, which means that geolocation error variances $\sigma_{coord}$ are small (as is discussed in section 3.4.1). It is hence reasonable in many cases to regard the geolocation errors in image 1 and image 2 as constant biases and to assume that $\sigma_{coord} = 0$ (section 3.4.2). When calculating the distance between two points with identical geolocation errors, we obtain hence $\sigma_d^2 = \sigma_{tr}^2$. Differences between the biases in image 1 and 2 affect the retrieval of ice drift. Deformation, on the other hand, is calculated from the relative change of size and shape of a given area between acquisitions of image 1 and image 2. The relative area change is independent of the regionally constant difference between the biases and depends only on the error variances (also here position uncertainties are assumed to be equal in image 1 and image 2). Therefore, deformation can be estimated with sufficient accuracy even if geolocation errors are large.

## 2.2 Uncertainty of drift velocity

The deformation is calculated from components of the velocity gradients according to Eq. (5). Hence, we have to consider the uncertainty in the measurements of velocity components $u_i$ and $v_i$. The components are calculated from $u = d_x/\Delta T$ and $v = d_y/\Delta T$, where $d_x = (x'-x)$ and $d_y = (y'-y)$ are the displacements in $x$- and $y$-direction, respectively, and $\Delta T$ is the time

interval needed for the position change from $(x, y)$ to $(x', y')$. Considering that errors in measuring time and positions are not correlated, we obtain from Eq. (2), taking into account a possible tracking error:

$$\sigma_u^2 = \frac{1}{\Delta T^2}\sigma_{d_x}^2 + \left(\frac{-d_x}{\Delta T^2}\right)^2 \sigma_{\Delta T}^2 = \frac{1}{\Delta T^2}(2\sigma_x^2 + \sigma_{tr_x}^2 + u^2\sigma_{\Delta T}^2) \qquad (6a)$$

$$\sigma_v^2 = \frac{1}{\Delta T^2}\sigma_{d_y}^2 + \left(\frac{-d_y}{\Delta T^2}\right)^2 \sigma_{\Delta T}^2 = \frac{1}{\Delta T^2}(2\sigma_y^2 + \sigma_{tr_y}^2 + v^2\sigma_{\Delta T}^2) \qquad (6b)$$

where $\sigma_{d_x}$, $\sigma_{d_y}$ are the uncertainties of the displacements (distances) in x- and y-direction, and $\sigma_{tr_x}$, $\sigma_{tr_y}$ are the corresponding components of the tracking error. If the uncertainty in timing, $\sigma_{\Delta T}^2$ is not zero, the assumption that $\sigma_u^2 = \sigma_v^2$ is only valid if $u^2 = v^2$. The uncertainty in speed $U$ (i. e. the magnitude of velocity vector $\mathbf{U}$) can be computed using Eq. (6), replacing $\sigma_u^2$ with $\sigma_U^2$, $\sigma_{d_x}^2$ with $\sigma_d^2$, $\sigma_{tr_x}^2$ with $\sigma_{tr}^2$, and $u$ with $U$, considering that $U = d/\Delta T = \sqrt[2]{u^2 + v^2}$, and $d = \sqrt[2]{(x' - x)^2 + (y' - y)^2}$. When calculating the relative error variance $\sigma_U/U$, one obtains Eq. (A1) in Hutchings et al. (2012).

If, on the other hand, both components of the vector $\mathbf{U}$ are determined separately (hence considering magnitude and direction), the result is different:

$$\sigma_U^2 = \left(\frac{\partial U}{\partial u}\right)^2 \sigma_u^2 + \left(\frac{\partial U}{\partial v}\right)^2 \sigma_v^2 = \left(\frac{u}{U}\right)^2 \sigma_u^2 + \left(\frac{v}{U}\right)^2 \sigma_v^2 \qquad (7)$$

Substituting Eq. (6) for $\sigma_u{}^2$ and $\sigma_v{}^2$ and setting $\sigma_{dx}{}^2 = \sigma_{dy}{}^2 = 2\sigma_{coord}^2 + \sigma_{tr}^2$ yields

$$\sigma_U^2 = \frac{2\sigma_{coord}^2 + \sigma_{tr}^2}{\Delta T^2} + \frac{\sigma_{\Delta T}^2}{\Delta T^2}\left(\frac{u^4 + v^4}{u^2 + v^2}\right) \qquad (8)$$

If $\sigma_{\Delta T}$ cannot be neglected, and if $u=0$ and $v=U$ or $v=0$ and $u=U$, the second term of Eq. (8) yields $U^2(\sigma_{\Delta T}^2/\Delta T^2)$, which is the uncertainty in speed given above. If, on the other hand, $u=v$ and hence $U^2=2u^2$, the second term is $0.5U^2(\sigma_{\Delta T}^2/\Delta T^2)$. This result may be viewed as if independent measurements of the two components $u$ and $v$ reduce the uncertainty contribution of $\sigma_{\Delta T}^2$.

## 2.3 Uncertainty of polygon area

The uncertainty of an area measurement is needed for application of Eq. (5) and equations presented in the following sections. The starting point for calculating the variance of error for the measurement of an area is the Surveyor's Area Formula valid for a polygon with an outline consisting of $n$ segments in a plane spanned by the $x$- and $y$-axis:

$$A = \frac{1}{2}\sum_{i=1}^{n}(x_i y_{i+1} - x_{i+1}y_i) \qquad (9)$$

Here $x_{n+1} \equiv x_1$ and $y_{n+1} \equiv y_1$ (closed polygon), $i$ is the summation index, and the boundary is traced in a counterclockwise sense. We have to consider that each coordinate appears twice in the sum of Eq. (9). When $i=k$ we have, e.g. for $x$: $x_k y_{k+1}$, and when $i=k-1$ we have $-x_k y_{k-1}$. For the law of error propagation, we need the derivatives:

$$\frac{\partial A}{\partial x_k} = \frac{1}{2}(y_{k+1} - y_{k-1}) \quad \text{and} \quad \frac{\partial A}{\partial y_k} = -\frac{1}{2}(x_{k+1} - x_{k-1}) \qquad (10)$$

where $k$ is the index of the derivative. Hence, we obtain

$$\sigma_A^2 = \frac{1}{4}\sum_{i=1}^{n}\left[\sigma_{i\_x}^2(y_{i+1} - y_{i-1})^2 + \sigma_{i\_y}^2(x_{i+1} - x_{i-1})^2\right] \tag{11}$$


We can assume that coordinate uncertainties $\sigma_{i\_x}^2 = \sigma_{i\_y}^2 = \sigma_{coord}^2$ are equal and the same for all measured positions. The uncertainty of the area is then

$$\sigma_A^2 = \frac{\sigma_{coord}^2}{4}\sum_{i=1}^{n}\left[(y_{i+1} - y_{i-1})^2 + (x_{i+1} - x_{i-1})^2\right] \tag{12}$$


Examples of applying Eq. (12) on basic polygons are shown in Fig. 2. Arbitrarily shaped triangles and quadrangles, which are basic patterns for arrays of three or four buoys and for grid cells in satellite images when applying the Lagrangian approach, are shown at the bottom. The $x$-$y$ coordinate system is here oriented such that the calculation of the uncertainty is easy. For any orientation of the triangle or quadrangle, side lengths and distances can be derived from the coordinates $(x, y)$ of the edge

points. For squares and equal-sided right-angled triangles, which are typical grid cells when retrieving ice drift from satellite images in a Eulerian approach, the uncertainty is directly proportional to the area. If a square grid cell is split into two triangles (as in Fig. 1), the uncertainty in area of each triangle is half that of the square.

For an assessment on how the polygon shape affects the magnitude of uncertainty we require that the enclosed area remains constant. The areas of a square with side length $L$ and a right-angled triangle with two sides of length $L_T$ are equal if $L_T = \sqrt{2}L$.

In this case we get $\sigma_A^2 = 2\sigma_{coord}^2 L^2$ for both square and triangle, which means that in this particular case the increase in number of vertices does not result in a decrease of $\sigma_A$. For a hexagon with $A = L^2$, on the other hand, one obtains $s^2 = 2L^2/3\sqrt{3}$ and $\sigma_A^2 = 1.44\sigma_{coord}^2 L^2$ (where $s$ is the length of a line segment on the boundary of the hexagon, see Fig.2). The issue of adding more vertices while keeping the shape of the polygon is addressed in Sect. 3.6.

The question arises how large the smallest detectable area change is in a SAR image? To address this question, we

assume a square grid cell with its vertices on the positions of adjacent displacement vectors and its sides parallel to the $x$- and $y$-axes of a Cartesian coordinate system. The cell covers $m \times m$ square-shaped pixels of side length $\Delta$x. The minimum possible change is to move one edge point by the side length of one pixel, either in $x$- or $y$- direction. This adds the area of a right triangle with legs $\Delta x$ and $m\Delta x$ ($\Delta y = \Delta x$) and the change of the area is $\Delta A = \frac{1}{2} m\Delta x^2$, i.e. $100/(2m)$ percent of the original area $(m\Delta x)^2$. Hence the larger the number of pixels in the area, the smaller the detectable relative area change. However, until now

we assumed that the position error is zero, but we have to consider the uncertainty of the area estimate, which is $\sigma_A^2 = 2\sigma_{coord}^2 m^2\Delta x^2$ for a square with $L = m\Delta x$. To be sure that a detected area change is real, $\Delta A$ needs to be larger than $\sigma_A$ or $\sigma_{coord}$ $< \frac{1}{2\sqrt{2}}\Delta x$.

## 2.4 Uncertainties for divergence, shear, vorticity, and total deformation in fixed grids

We consider a grid with displacement or drift velocity vectors on the vertices. For calculating the deformation parameters, we

need the velocity gradients $u_x, u_y, v_x, v_y$, obtained from Eq. (5). Formally, the gradients depend on the area A, positions $(x_i, y_i)$, and velocities $(u_i, v_i)$, see Sect. 2.5. Here we assume that the geo-referencing of the satellite images is accurate. In this case, the positions $(x_i, y_i)$ of vertices and the area of each grid cell are known precisely, which means that $\sigma_{coord} = 0$ and $\sigma_A = 0$. The displacement or velocity vectors, however, have an uncertainty related to the tracking error. With $\partial u_x/\partial u_k = (y_{k+1} - y_{k-1})/2A$ and again considering that two terms in the sum Eq. (5) include $u_i$, the uncertainty of the velocity gradient in the x-direction is

(Griebel and Dierking, 2018):

$$\sigma_{ux}^2 = \frac{\sigma_u^2}{4A^2} \sum_{i=1}^n (y_{i+1} - y_{i-1})^2 \tag{13}$$

and analogous equations for the other gradient components. The divergence is $\dot{\varepsilon}_{div} = u_x + v_y$, Eq. (3a), and the corresponding uncertainty is $\sigma_{div} = \sqrt[2]{\sigma_{ux}^2 + \sigma_{vy}^2}$, if $u_x$ and $v_y$ are independent. Throughout this section we assume that $\sigma_U = \sigma_u = \sigma_v$ and $\sigma_{\Delta T} = 0$, hence the error variance for the divergence is

$$\sigma_{div}^2 = \frac{\sigma_U^2}{4A^2} \sum_{i=1}^n [(y_{i+1} - y_{i-1})^2 + (x_{i+1} - x_{i-1})^2] = \frac{\sigma_{tr}^2}{4A^2 \Delta T^2} \sum_{i=1}^n [(y_{i+1} - y_{i-1})^2 + (x_{i+1} - x_{i-1})^2] \tag{14}$$

Equation (14) resembles the uncertainty for a polygon, Eq. (11). Since the position uncertainty $\sigma_{\text{coord}}$ is set to zero, the uncertainty of velocity $U$ is only a function of the tracking uncertainty $\sigma_{tr}$, see Eq. (8) (assuming $\sigma_{\Delta T} = 0$). For the vorticity Eq. (3b) one obtains $\sigma_{vrt} = \sqrt[2]{\sigma_{vx}^2 + \sigma_{uy}^2}$ and thus the same expression as for the divergence. The shear rate is given by Eq. (3c). Calculating the derivatives with respect to the velocity gradient components and applying the law of error propagation yields:

$$\sigma_{shr}^2 = \frac{(u_x - v_y)^2}{\dot{\varepsilon}_{shr}^2} (\sigma_{ux}^2 + \sigma_{vy}^2) + \frac{(u_y + v_x)^2}{\dot{\varepsilon}_{shr}^2} (\sigma_{uy}^2 + \sigma_{vx}^2) \tag{15a}$$

With $\phi = \frac{1}{2} \arctan((u_y + v_x) / (u_x - v_y))$, which gives the principal direction of shear, and using Eq. (3c) and relations $cos^2(\arctan(x)) = 1/(1 + x^2)$ and $sin^2(\arctan(x)) = x^2/(1 + x^2)$, Eq. (15a) can be expressed as

$$\sigma_{shr}^2 = cos^2(2\phi) \, \sigma_{div}^2 + sin^2(2\phi) \, \sigma_{vrt}^2 \tag{15b}$$

Since $\sigma_{div}^2 = \sigma_{vrt}^2$ and $cos^2(2\phi) + sin^2(2\phi) = 1$, the error variances are equal for divergence, vorticity and shear. For the total deformation, Eq. (3d), we need the derivatives $\partial(\dot{\varepsilon}_{tot})/\partial(\dot{\varepsilon}_{shr})$ and $\partial(\dot{\varepsilon}_{tot})/\partial(\dot{\varepsilon}_{div})$ with which we obtain

$$\sigma_{\text{tot}}^2 = \frac{\dot{\varepsilon}_{shr}^2}{\dot{\varepsilon}_{tot}^2} \sigma_{shr}^2 + \frac{\dot{\varepsilon}_{div}^2}{\dot{\varepsilon}_{tot}^2} \sigma_{div}^2 \tag{16a}$$

If we define $\theta = \arctan(\varepsilon_{shr} / \varepsilon_{div})$ (Stern et al., 1995), Eq. (16a) can be rewritten as

$$\sigma_{\text{tot}}^2 = sin^2(\theta) \, \sigma_{shr}^2 + cos^2(\theta) \, \sigma_{div}^2 \tag{16b}$$

The angle $\theta$ gives the relative contributions of divergence and shear: pure divergence is $\theta = 0°$, uniaxial extension is $\theta = 45°$, pure shear is $\theta = 90°$, uniaxial compression is $\theta = 135°$, and pure convergence is $\theta = 180°$. Since the uncertainties for shear and divergence are of equal magnitude, it follows that

$$\sigma_{tot}^2 = \sigma_{shr}^2 = \sigma_{div}^2 = \sigma_{vrt}^2 \tag{17}$$

In the following, we assume that $\sigma_{\Delta T}$ can be neglected and that the standard deviations for the velocity components $u$ and $v$ are equal. Using Eq. (14) for a square cell, we obtain for the uncertainty of the divergence:

$$\sigma_{\text{div}}^2 = \frac{\sigma_U^2}{4A^2} (4L^2 + 4L^2) = \frac{2\sigma_d^2}{L^2 \Delta T^2} = \frac{2\sigma_{tr}^2}{L^2 \Delta T^2} \tag{18}$$

with $A = L^2$, $\sigma_U^2 = \sigma_d^2/\Delta T^2$, and $\Delta T = t-t_0$ as above. Since the position uncertainty is zero in the case investigated here, $\sigma_d^2$ (which equals $2\sigma_{coord}^2 + \sigma_{tr}^2$, see Section 2.1) depends only on the tracking error (compare to Eq. (17) in Lindsay and Stern, 2003).

## 2.5 Uncertainties of deformation parameters, general case

For an array of buoys, we have to consider errors of the area, the buoy velocity components $u$ and $v$, and the coordinates $(x, y)$ of each buoy position. The general case does also apply to SAR images if geolocation error variances cannot be neglected. A buoy array consists of single buoys arbitrarily positioned over a plane. When connecting all buoy positions with lines, a polygon of area A is formed in which distances between adjacent buoys are usually different. The starting point is Eq. (5). In the following equations summation bounds from $i = 1$ to $n$ are omitted. We note that the equations in this section have been independently derived by Bouchat and Tremblay (2020) as well.

For the uncertainty in $u_x$ we obtain

$$\sigma_{ux}^2 = \sigma_A^2 \left(\frac{\partial u_x}{\partial A}\right)^2 + \sigma_u^2 \sum \left(\frac{\partial u_x}{\partial u_i}\right)^2 + \sigma_y^2 \sum \left(\frac{\partial u_x}{\partial y_i}\right)^2 \tag{19}$$

With $\frac{\partial u_x}{\partial A} = -\frac{1}{2A^2}\sum(u_{i+1} + u_i)(y_{i+1} - y_i)$,

$\frac{\partial u_x}{\partial u_k} = \frac{1}{2A}(y_{k+1} - y_{k-1})$, and $\frac{\partial u_x}{\partial y_k} = -\frac{1}{2A}(u_{k+1} - u_{k-1})$, Eq. (19) reads:

$$\sigma_{ux}^2 = \frac{\sigma_A^2}{4A^4}[\sum(u_{i+1} + u_i)(y_{i+1} - y_i)]^2 + \frac{\sigma_u^2}{4A^2}\sum(y_{i+1} - y_{i-1})^2 + \frac{\sigma_y^2}{4A^2}\sum(u_{i+1} - u_{i-1})^2 \tag{20}$$

The first term on the right side is calculated on line segments connecting adjacent vertices $(i+1, j+1)$, $(i, j)$, the second and third on chords from $(i+1, j+1)$ to $(i-1, j-1)$. Assuming $\sigma_{coord}^2 = \sigma_x^2 = \sigma_y^2$; $\sigma_U^2 = \sigma_u^2 = \sigma_v^2$ (the latter follows from $\sigma_T^2/\Delta T^2 \approx 0$) one obtains for the divergence:

$$\sigma_{div}^2 = \sigma_{ux}^2 + \sigma_{vy}^2 = \frac{\sigma_A^2}{4A^4}\{[\sum(u_{i+1} + u_i)(y_{i+1} - y_i)]^2 + [\sum(v_{i+1} + v_i)(x_{i+1} - x_i)]^2\}$$
$$+ \frac{\sigma_U^2}{4A^2}[\sum(x_{i+1} - x_{i-1})^2 + \sum(y_{i+1} - y_{i-1})^2] + \frac{\sigma_{coord}^2}{4A^2}[\sum(u_{i+1} - u_{i-1})^2 + \sum(v_{i+1} - v_{i-1})^2] \tag{21}$$

where the first term can be written as $\frac{\sigma_A^2(u_x^2 + v_y^2)}{A^2}$, considering Eq. (5). For the vorticity, only the first term is different:

$$\sigma_{vrt}^2 = \sigma_{uy}^2 + \sigma_{vx}^2 = \frac{\sigma_A^2}{4A^4}\{[\sum(u_{i+1} + u_i)(x_{i+1} - x_i)]^2 + [\sum(v_{i+1} + v_i)(y_{i+1} - y_i)]^2\}$$
$$+ \frac{\sigma_U^2}{4A^2}[\sum(x_{i+1} - x_{i-1})^2 + \sum(y_{i+1} - y_{i-1})^2] + \frac{\sigma_{coord}^2}{4A^2}[\sum(u_{i+1} - u_{i-1})^2 + \sum(v_{i+1} - v_{i-1})^2] \tag{22}$$

Here, the first term can be written as $\frac{\sigma_A^2(u_y^2 + v_x^2)}{A^2}$. The first terms in Eqs. (21) and (22), right side, consider that the relative error variance of the area affects the magnitude of the average velocity gradients. The second term is the variance of divergence/vorticity of the velocity field in a fixed grid where positions of vertices are known precisely, Eq. (14). The last term takes into account the effect of uncertainties in the positions of buoys in the field of velocity vectors. The velocity is usually determined from buoy positions separated by a time interval $\Delta T = T_{i+1} - T_i$. However, within $\Delta T$ also the buoy array changes its area and shape. Hence an alternative approach would be to determine the average velocity from positions at $T_{i-1}$,

$T_i$, and $T_{i+1}$ and link it with the geometric properties of the buoy array at time $T_i$. For the shear and total deformation, the results

are formally equal to Eqs. (15a), (15b) and (16a), (16b), where now $\sigma_{ux}$, $\sigma_{uy}$, $\sigma_{vx}$, $\sigma_{vy}$ are calculated using Eq. (20) and analogous expressions. Note that in this case the uncertainties of divergence, vorticity, shear, and total deformation differ from one another, unless $\sigma_{ux}^2 = \sigma_{uy}^2 = \sigma_{vx}^2 = \sigma_{vy}^2$. In practical applications, they can be evaluated numerically. This requires the knowledge of uncertainties $\sigma_{coord}$ for buoys. and $\sigma_{coord}$, $\sigma_{tr}$ for satellite images.

## 3 Discussion

Eqs. (21) and (22) together with Eq. (15) and (16) provided above indicate that statistical uncertainties are not only influenced by geolocation and tracking errors but also depend on the shape and size of grid cells and buoy arrays. In the following discussion we consider magnitudes of geolocation and tracking errors reported in the literature and selected squares and triangles as examples for grid cells in SAR images (Lindsay, 2002; Bouillon and Rampal, 2015) and for splitting large buoy arrays into smaller units (Hutchings et al., 2012; Itkin et al., 2017). The effect of combining several cells is investigated.

Finally, we focus on the range of validity of the equations derived in Sect. 2, and alternative methods of analysis.

### 3.1 Typical magnitudes of deformation parameters

The statistical uncertainties have to be related to the typical magnitudes of the deformation parameters. According to Leppäranta (2011, p.70) the total deformation of drifting ice typically varies between around 90% d$^{-1}$ in the marginal ice zone to 0.9% d$^{-1}$ in the central Arctic. For the vorticity, magnitudes up to 9% d$^{-1}$ = ½ (0.09) revolutions d$^{-1}$ = 16.2° d$^{-1}$ were observed.

Hutchings et al. (2012, Figs. 4 and 7) analyzed displacements of an array of 24 buoys deployed in the Weddell Sea on first- and second-year ice with concentrations above 90 percent. For divergence, they found most values between -90% d$^{-1}$ and 90% d$^{-1}$ at a spatial scale of 10 km; at 60 km scale mainly between ±25% d$^{-1}$, and up to 35% d$^{-1}$ for the shear. Note that spatial scales are mentioned here since they affect the observed magnitudes of deformation (e.g. Marsan et al., 2004). Itkin et al. (2017) observed exceptional events of strong divergence and shear of up to 200% d$^{-1}$ from buoys in an area north of Svalbard (their

Fig. 4), but over most of the measurement period, magnitudes were lower. At scales of 15 km or less, values for divergence covered the range ±20% d$^{-1}$ over several days to weeks, but also variations of about ±100% d$^{-1}$ occurred for three weeks. Shear was close to zero for a few days but varied mainly from 20 to 70% d$^{-1}$ for three weeks. At measurement scales larger than 60 km, the magnitudes of divergence and shear were lower than at ≤15 km scale, with the exception of very short periods during which the opposite was the case. Magnitudes for divergence were roughly at ±10% d$^{-1}$ with occasional minima and

maxima in the range of ±100% d$^{-1}$, and for shear most values were ≤10% d$^{-1}$ with a few peaks at about 100% d$^{-1}$. Based on merged velocity measurements from buoys and different satellite sensors, Lindsay (2002) provided a table for monthly averaged values of divergence (-0.6 to 0.5% d$^{-1}$), shear (0.9 to 4% d$^{-1}$), and vorticity (-2.3 to 3.2% d$^{-1}$) from the Beaufort Sea at a scale of 100 km. Stern and Moritz (2002, Fig. 4) used SAR images and found decreasing magnitudes for the divergence for increasing spatial scales from 50×50 km to 200×200 km in the Beaufort Sea. Magnitudes were largest between August and

February with minima/maxima between -5% d$^{-1}$ and 5% d$^{-1}$ at a scale of 50 km, decreasing at larger scales. Note that the uncertainties resulting from the equations given in the subsections below have to be multiplied by 100 to obtain a value in percent per time unit.

### 3.2 Uncertainties for areas of simple geometric shape

In general, the uncertainty of the deformation parameters depends on the ratio $\sigma_{coord}^2/A^2$ (since $\sigma_A$ and $\sigma_U$ are functions

of $\sigma_{coord}$), hence for given geolocation and tracking errors it decreases with increasing area. The first term in Eqs. (21) and (22) is smallest if, for given area and velocity gradients, $\sigma_A$ is at a minimum. For an arbitrary triangle with sides $a$, $b$, $c$, the

uncertainty $\sigma_A{}^2$ is $0.25\sigma_{coord}{}^2\,(a^2+b^2+c^2)$ (see Fig. 2). Of all triangles with the same base and the same area $A$, the equal-sided triangle with $a = b = c$ has the smallest perimeter and hence the lowest uncertainty, which is $\sigma_A^2 = \sqrt{3}\sigma_{coord}^2 = 1.73\sigma_{coord}^2$ for a unit area. (This follows from the equations for the area of the equal-sided triangle which is $A = \frac{\sqrt{3}}{4}a^2$ and for the uncertainty $\sigma_A^2 = \frac{3a^2}{4}\sigma_{coord}^2$ if $A = 1$). In case of rectangles and rhombi, squares have the smallest perimeter (see Fig. 2). In both cases the uncertainty is $\sigma_A^2 = 2\sigma_{coord}^2$ for a unit area, hence larger than for the equal-sided triangle. For the regular hexagon, which is composed of six equal-sided triangles, one obtains $\sigma_A^2 = \frac{5}{2\sqrt{3}}\sigma_{coord}^2 = 1.44\sigma_{coord}^2$ (Fig. 2). So the progression of $\sigma^2{}_A/\sigma^2{}_{coord}$ from triangles to squares to hexagons goes from $1.73A$ to $2.00A$ to $1.44A$.

### 3.3 Uncertainties in time

The accuracy of time readings for the acquisitions of satellite images is on the order of sub-seconds. The product of sea ice drift velocity and uncertainty of time reading appears on the right-hand side of Eq. (6): $2\sigma_{coord}{}^2 + \sigma_{tr}{}^2 + u^2\,\sigma_{\Delta T}{}^2$. Average sea ice drift velocities range mostly from 0 to 0.35 m/s (Rampal et al., 2009). Kræmer et al. (2015) determined instantaneous line-of-sight ice drift velocities, using Doppler frequency measurements from SAR, and found values as large as 0.4-0.6 m/s. If we assume a maximum value of u = 1 m/s and a maximum uncertainty of time readings of 1 millisecond, the term $u^2\,\sigma_{\Delta T}{}^2$ on the right side of Eq. (6) is $10^{-6}$ m$^2$ at the most. It can be neglected compared to the typical values of terms $\sigma_x{}^2$, $\sigma_y{}^2$ and $\sigma_{tr_x}{}^2$, $\sigma_{tr_y}{}^2$ in Eq. (6) (see Sect. 3.4 for a discussion of the effect of position and tracking errors). The uncertainty $\sigma_{\Delta T}$ of the GPS time (used both for buoys and satellites such as Sentinel-1) is given as better than one millisecond (see, e.g. websites [1] and [2]). Similar considerations apply to Eq. (8). Hence, in Eqs. (21) and (22) we have $\sigma_U{}^2 = (2\sigma_{coord}{}^2 + \sigma_{tr}{}^2)/\Delta T^2$ both for velocity retrievals from satellite image pairs and buoy arrays. For given position and tracking errors, the second term in Eq. (21) decreases with increasing time interval $\Delta T$ and area $A$. The third term involving the coordinate uncertainty $\sigma_{coord}$ also decreases with increasing area $A$.

Another issue that has to be considered is the time synchronization between individual buoys in an array. Differences of a few seconds may be possible in practice. In the following discussion we assume that position data of all buoys are exactly synchronized but also discuss an example for which this was not the case in Section 3.5.

### 3.4 Deformation retrievals from square grid cells

Here we first focus on the retrieval of deformation parameters calculated from square grid cells in SAR images or from square-shaped buoy arrays. For SAR images, we consider the case in which geolocation errors may have slight variations, hence $\sigma_{coord} \neq 0$. If a square of side length $L$, with sides parallel to the $x$- and $y$-axes, is positioned in a spatially varying velocity field as shown in Fig. 3, the uncertainty of the divergence is:

$$\sigma_{div}^2 = \frac{3\sigma_{coord}^2}{L^2}\left(u_x^2 + v_y^2\right) + \frac{\sigma_{coord}^2}{L^2}\left(u_y^2 + v_x^2\right) + \frac{4\sigma_{coord}^2}{\Delta T^2 L^2} + \frac{2\sigma_{tr}^2}{\Delta T^2 L^2} \tag{23}$$

This follows from Eq. (21) with the velocities given in Fig. 3 at the edges 1-4 of the square. The uncertainty of the vorticity is from Eq. (22)

$$\sigma_{vrt}^2 = \frac{3\sigma_{coord}^2}{L^2}\left(u_y^2 + v_x^2\right) + \frac{\sigma_{coord}^2}{L^2}\left(u_x^2 + v_y^2\right) + \frac{4\sigma_{coord}^2}{\Delta T^2 L^2} + \frac{2\sigma_{tr}^2}{\Delta T^2 L^2} \tag{24}$$

Uncertainties of shear and total deformation can be calculated using Eqs. (15b) and (16b) as weighted averages of the error variances of divergence and vorticity, and of shear and divergence, respectively. The second term in Eq. (23) and first

term in Eq. (24) indicates that the uncertainties of divergence and vorticity are affected by contributions from pure shear. The third and fourth term of Eq. (23) are independent of the velocity gradients and are only a function of position and tracking error, time interval between position measurements, and size of the square. The fourth term is equal to Eq. (17) in Lindsay and Stern (2003). In general, it is more realistic to assume that arrays of four buoys are arbitrarily shaped quadrangles. As mentioned in Section 1, drift vectors from SAR image pairs are irregularly spaced if calculated using feature tracking (e.g.

Komarov and Barber, 2014; Muckenhuber et al., 2016; Demchev et al., 2017). While $\sigma_{tr}$, $\sigma_{coord}$, and $\Delta T$ are constant, $\sigma_A$ and $A$ depend on the size and shape of the quadrangle that changes from grid cell to grid cell (Figs. 1c and 2). In this case the most convenient approach for calculating deformation parameters is the application of Eqs. (21) and (22) together with Eqs. (15) and (16). We emphasize, however, that the heterogeneous spatial distribution of drift vectors is regarded as a disadvantage for evaluating and analyzing sea ice deformation, since the latter is a scale-dependent process (Korosov and Rampal, 2017).

### 3.4.1 Geolocation error and uncertainties in SAR images


    When ice drift is retrieved from images of modern SAR systems, the contribution of those terms that depend on $\sigma_{coord}/L$ can usually be neglected, as we will show below. For Envisat ASAR stripmap and wide-swath mode images (IM and WSM), e.g., Small et al. (2005) reported differences between measured positions of reflectors and their positions in the SAR image of 1.63±0.82 m in azimuth (considering bi-static correction) and 2.02±0.51 m in slant range for normal imaging mode

in single-look complex format. Ground range products require the transformation from slant- to ground-range as an additional step. When judging the effect of position errors on the uncertainty of divergence and vorticity, the systematic bias (mean error) of positions affects all vertices of a grid cell in the same way, hence only the standard deviation $\sigma$ has to be considered as geolocation uncertainty. Considering the $\sigma$-values of position errors given above, we use a value of 1 m as a conservative estimate of the azimuth and ground-range position uncertainty for IM. For ground-range WSM images, the accuracy of

positioning was better than one pixel. If we assume that the ratio $\sigma[m]/\sigma[pixel]$ is approximately same for IM and WSM, the uncertainty for the latter is about 7 m at maximum. In the study of Hollands and Dierking (2011), e.g., resolution pyramids and cascades are used for retrieving sea ice displacements from Envisat ASAR IM and WSM data. For the level of highest spatial resolution, the side lengths of the grid cells (distance between adjacent displacement vectors) was 300 m for IM and 1200 m for WSM. Hence, the corresponding ratios $\sigma_{coord}{}^2/L^2$ are on the order of $1^2/300^2 \approx 10^{-5}$ and $7^2/1200^2 \approx 3.4 \times 10^{-5}$,

respectively. For modern SAR systems such as TerraSAR-X and Sentinel-1, the positioning accuracy is even better than for Envisat (e.g. Schubert et al., 2008; Schubert et al., 2017). The geolocation error of older SAR systems, however, is larger. In their analysis of drift and deformation products from the RADARSAT Geophysical Processor System (RGPS), Lindsay and Stern (2003) report geolocation errors (to be treated as bias, see above) of 225 m and 277 m for RADARSAT ScanSAR images. For the combined geolocation and tracking uncertainty $\varepsilon_{RGPS} = \sqrt{2\sigma_{coord}^2 + \sigma_{tr}^2}$ they found a value of 286 m. With a tracking

uncertainty of 100 m, the geolocation uncertainty is hence 190 m. The initial grid cells used for the RGPS are squares of 10 km side length, but they change their shape in successive time steps since the RGPS drift and deformation products are based on the Lagrangian approach. The ratio $\sigma_{coord}{}^2/L^2$ is approximately $200^2/10000^2 = 4.0 \times 10^{-4}$. The third and fourth term in Eqs. (23) and (24) can be directly computed from position and tracking error, the time interval $\Delta T$ between image acquisitions, and the grid cell size. The ratio between the fourth and the third term is $\sigma_{tr}{}^2/2\sigma_{coord}{}^2$. In the following section, the relative

contribution of single terms in Eqs. (23) and (24) are discussed.

### 3.4.2 Examples: Uncertainties versus true magnitudes of deformation

    According to Sect. 3.1, a value of $\pm 1$ d$^{-1}$ can be regarded as large divergence rates which is rarely exceeded in reality. Large values of shear were at about 0.7 d$^{-1}$. Considering the numbers for divergence and shear given in Sect. 3.1 we can deduce that the terms $\left(u_x^2 + v_y^2\right)$ and $\left(u_y^2 + v_x^2\right)$ in Eqs. (23) and (24) are $< 1$ d$^{-2}$ in most cases, and at larger length scales and weak

deformation more likely on the order of $10^{-1}$ d$^{-2}$ or $10^{-2}$ d$^{-2}$. This means that $\sigma_{coord}{}^2/L^2$ and $3\sigma_{coord}{}^2/L^2$ can be used as upper bounds for the first and second term in Eqs. (23) and (24) (see Table 1).

Hollands and Dierking (2011) found tracking errors between 0.8 and 1.6 pixels (their Tables 3 and 4, standard deviations), which corresponded to 20 - 40 m for IM (pixel size 25 m) and 120 – 240 m for WSM (pixel size 150 m). With $\sigma_{coord}$ = 1 m for IM and 7 m for WSM, the ratios between fourth and third term in Eqs. (23) and (24) are hence 200 – 800 for

IM and 147 – 588 for WSM. In this case the first three terms can be neglected compared to the fourth (see Table 1, columns 2 and 3, in which the range from minimum to maximum values for the fourth term is estimated using corresponding combinations of $\Delta T$ and $\sigma_{tr}$). With a grid cell size of $L$ = 300 m (IM) and 1200 m (WS), and time differences ranging from 1.2 to 5.8 days for IM image pairs and from 2 to 6 days for WSM image pairs, the uncertainties $\sigma_{div}$ and $\sigma_{vrt}$ were between 2.4% d$^{-1}$ and 14 % d$^{-1}$ for IM and 3.5% d$^{-1}$ and 12.7% d$^{-1}$ for WSM (calculated for each image pair listed in Table 1 of Hollands and Dierking

(2011), with the corresponding tracking errors from their Tables 3 and 4). Comparing these values to the typical magnitudes of divergence and vorticity in Sect. 3.1, the respective uncertainties are too large in areas of weaker deformation.

Lindsay and Stern (2003) calculated deformation parameters for the RGPS initial velocity grid ($L$=10 km), and a time interval $\Delta T$ of 3 days. They use a tracking error of 100 m for RADARSAT ScanSAR images (pixel size 100 m) and assumed that the geolocation error can be regarded as bias with zero uncertainty. Hence, only the fourth term of Eqs. (23) and

(24) is used (their Eq. (17)), and uncertainties for divergence and vorticity are 0.5% d$^{-1}$ (Table 1, column 4). However, when considering the uncertainty of the geolocation error mentioned in Sect. 3.4.1, the fourth term contributes less than the other three terms (Table 1, column 5). Only if terms $(u_x^2 + v_y^2)$ and $(u_y^2 + v_x^2)$ are of magnitudes < 0.001 d$^{-2}$, the first and second term can be neglected compared to the third term.

At first sight, larger time intervals and grid cells seem to be advantageous to keep the uncertainties of deformation

parameters at a low level. However, larger time intervals may cause problems in the retrieval of the ice drift field, since ice structures, which serve as reference for the retrieval, may change or even vanish with time. Larger grid cells may smooth out local variations of deformation.

If the first and second term in Eq. (23) and (24) can be neglected, i.e. when magnitudes of deformation parameters are low (which is most likely for measurements over larger spatial scales and for weak deformation events), we can determine

the minimum grid cell size that is required to keep the uncertainties of divergence and vorticity below a given threshold. If we assume an uncertainty threshold of 1% d$^{-1}$, then the third and fourth term of Eqs. (23) and (24) tells us that the ratio between combined position and tracking uncertainty and grid cell size should satisfy $\sqrt{4\sigma_{coord}^2 + 2\sigma_{tr}^2}/L \leq 0.01\ [d^{-1}] \times \Delta T\ [d]$. If $\sigma_{coord} \ll \sigma_{tr}$ we obtain $\sigma_{tr}/L \leq 0.01\ [d^{-1}] \times \Delta T\ [d]/\sqrt{2} \cong 0.007\ [d^{-1}] \times \Delta T\ [d]$. For $\Delta T$ = 1 d, this means a grid cell length of roughly $150 \times \sigma_{tr}$ (uncertainty 1% d$^{-1}$) or larger (uncertainty < 1% d$^{-1}$).

**3.5 Deformation retrievals from triangular grid cells or buoy arrays**

Also triangles are used for calculations of deformation parameters in SAR images (e.g. Bouillon and Rampal, 2015; Griebel and Dierking, 2018) and they form the smallest units of buoy arrays (e.g. Hutchings et al., 2011; Hutchings et al., 2012). Using the same approach as for the square above, we obtain for a triangle with its base $a$ parallel to the $x$-axis (Fig. 4):

$$\sigma_{div}^2 = \frac{\sigma_{coord}^2(a^2+b^2+c^2)}{h_a^2 a^2}\left(u_x^2 + v_y^2\right) + \frac{(2\sigma_{coord}^2 + \sigma_{tr}^2)(a^2+b^2+c^2)}{\Delta T^2 h_a^2 a^2}$$

$$+ \frac{2\sigma_{coord}^2}{h_a^2 a^2}\left[(u_x^2 + v_x^2)(a^2 + a_1^2 - aa_1) + (u_y^2 + v_y^2)h_a^2 + (u_x u_y + v_x v_y)(2a_1 - a)h_a\right] \tag{25}$$

Sides $b$, $c$, height $h_a$, and segments $a_1$, $a_2$ are shown in Fig. 4. For the vorticity, the sum $(u_x^2 + v_y^2)$ in the first term has to be replaced by $(u_y^2 + v_x^2)$. Equation (25), which is shown here for an acute triangle (all internal angles <90°), is also valid for an obtuse triangle (one internal angle >90°) setting $a_1$ negative and $a_2$ to zero. For a right triangle with $b = a$, $c = \sqrt{2}a$, $h_a = a$, and $a_1 = a$, Eq. (25) yields

$$\sigma_{div}^2 = \frac{6\sigma_{coord}^2}{a^2}\left(u_x^2 + v_y^2\right) + \frac{2\sigma_{coord}^2}{a^2}\left(u_y^2 + v_x^2 + \left(u_x u_y + v_x v_y\right)\right) + \frac{8\sigma_{coord}^2}{\Delta T^2 a^2} + \frac{4\sigma_{tr}^2}{\Delta T^2 a^2} \tag{26a}$$

However, if the right angle is placed at the left side of the triangle, i.e. $c = a$, $b = \sqrt{2}a$, $h_a = a$, and $a_1 = 0$, the resulting equation changes to:

$$\sigma_{div}^2 = \frac{6\sigma_{coord}^2}{a^2}\left(u_x^2 + v_y^2\right) + \frac{2\sigma_{coord}^2}{a^2}\left[u_y^2 + v_x^2 - \left(u_x u_y + v_x v_y\right)\right] + \frac{8\sigma_{coord}^2}{\Delta T^2 a^2} + \frac{4\sigma_{tr}^2}{\Delta T^2 a^2} \tag{26b}$$

Similarly as for the grid of squares, the contributions of terms 1-3 of Eqs. (26a) and (26b) can be neglected when geolocation uncertainties are much smaller than tracking uncertainties. When comparing the third and fourth terms of Eqs. (26) and (23) one finds that the squared uncertainty of a right triangle is two times the squared uncertainty of a square for $a = L$ and identical $\sigma_{coord}$, $\sigma_{tr}$, and $\Delta T$, which can be attributed to the reduced coverage of the triangle over the varying velocity field. For an uncertainty of 1% d$^{-1}$, we obtain a value of $\leq 0.005$ [d$^{-1}$] $\times \Delta T$ [d] for the ratio $\sigma_{tr}/a$, corresponding to a base length $a$ of $200 \times \sigma_{tr}$ if $\Delta T = 1$ d.

The uncertainty of the equal-sided triangle ($c = b = a$, $h_a^2 = 3a^2/4$, and $a_1 = a/2$) is

$$\sigma_{div}^2 = \frac{6\sigma_{coord}^2}{a^2}\left(u_x^2 + v_y^2\right) + \frac{2\sigma_{coord}^2}{a^2}\left(u_y^2 + v_x^2\right) + \frac{8\sigma_{coord}^2}{\Delta T^2 a^2} + \frac{4\sigma_{tr}^2}{\Delta T^2 a^2} \tag{27}$$

Note that compared to a square of length $L$, the area of an equal-sided triangle with base $L$ is $0.433A_{square}$. The area of an arbitrary triangle with constant base increases when changing its shape from the equal-sided to the right triangle.

### 3.5.1 Uncertainties in position and temporal sampling

For buoys, the tracking error is zero. Itkin et al. (2017) quoted 25 m as geolocation accuracy for stationary buoys but used 50 m to account for effects of buoy drift. One of us (Hutchings) analyzed the position errors of GPS receivers in the Fairbanks (Alaska) region. The errors were normally distributed for position data collected at the same location for several days. The relative position error between pairs of GPS receivers, which has to be used for deformation calculations, was 2 m over distances of 1-10 km. Reported time intervals between acquisitions of buoy positions range from 10 seconds to 3 hours (Hutchings, 2012; Itkin et al., 2017) with uncertainties in time less then milliseconds (see above). Hutchings at al. (2012), however, mention also a time error of 30 seconds, which was due to the acquisition times of the buoys not being exactly time coincident. In such exceptional case, the second term on the right-hand side of Eq. (8) may have to be considered. If the ice drifts in $x$-direction (i.e. $v = 0$), the right-hand side of Eq. (8) reads $(2\sigma_{coord}^2 + u^2 \sigma_{\Delta T}^2) / \Delta T^2$ ($\sigma_{tr} = 0$ for buoys). Here we ask: What is the maximum value of the drift velocity for which the term $u^2 \sigma_{\Delta T}^2$ can still be neglected? Our criterion for neglecting it is that its value is 1% of $2\sigma_{coord}^2$ or less. Then the velocity $u$ must be equal or smaller than 424 m/h if we assume that $\sigma_{coord} = 25$ m and $\sigma_{\Delta T} = 30$ s. If $\sigma_{coord} = 2$ m, the result is $u = 34$ m/h. The speed of sea ice drift ranges mainly between 0 and 1.3 km/h, with possible extreme values around 3.6 km/h (see Sect. 3.3) which means that the term $u^2\sigma_{\Delta T}^2$ has to be taken into account in most cases. Conversely, we may ask how large the acceptable maximum temporal sampling error is so that the

second term is negligible (i.e. < 1% compared to the first term). With $u_{max}$ = 3.6 km/h = 1 m/s and $\sigma_{coord}$ = 2 m one obtains $\sigma_{\Delta T}$

= 0.3 s, and for $\sigma_{coord}$ = 25 m it is $\sigma_{\Delta T}$ = 3.5 s.

### 3.5.2 Optimal sizes of buoy arrays

In this section we ask how large the area of a triangle-shaped buoy array has to be chosen to keep the uncertainty for deformation below a given threshold? We assume that the temporal sampling error can be neglected. The time interval $\Delta T$ is set to the temporal sampling rate of buoy positions. For buoy arrays, the tracking error is zero. With a given threshold for

divergence, e.g., one can use Eqs. (26) and (27) to calculate base $a$ of right-angle or equal sided triangles. Solutions of these simple cases can serve for approximately fixing the optimal area size for triangles of arbitrary shapes. For such triangles, the corresponding Eq. (25) cannot be directly solved since they need to be described by additional geometric parameters besides base $a$.

The first two terms of Eqs. (26) and (27) require the knowledge of the sea ice velocity field and its gradients. We

will here focus on cases for which these terms can be neglected. This requires that $8\Delta T^{-2} \gg 6$, i.e. that $\Delta T$ is small. Itkin et al. (2017) analyzed deformation for constellations of three buoys using temporal sampling intervals of $\Delta T_1$ = 1 h and $\Delta T_2$ = 3 h, which results in $\Delta T_1^{-2}$ = 576 d$^{-2}$ and $\Delta T_2^{-2}$ = 64 d$^{-2}$. For a large fraction of measured divergence and shear data we can assume that $(u_x^2 + v_y^2)$ and $(u_y^2 + v_x^2)$ are smaller than one (see Sect. 3.4.2) and neglect the first two terms in Eqs. (26) and (27). At low magnitudes of deformation this is also justified for $\Delta T_3$ = 24 h, which gives $\Delta T_3^{-2}$ = 1 d$^{-2}$.

Using only the third term $(8/\Delta T^2) \times (\sigma_{coord}^2/a^2)$ the uncertainty of the divergence can be expressed as $\sigma_{div}$ = 71/$a$ h$^{-1}$ for $\Delta T_1$ = 1 h and 24/$a$ h$^{-1}$ for $\Delta T_2$ = 3 h, where $\sigma_{coord}$ = 25 m and the value for base $a$ has to be given in meters. In the following we accept an uncertainty of 5% or less relative to the majority of the magnitudes of divergence derived in Itkin et al., 2017 which are ≤ 0.4 d$^{-1}$ = 0.017 h$^{-1}$. Hence the uncertainty is $\sigma_{div}$ =0.00085 h$^{-1}$, which means that base $a$ of the triangle has to be larger than 83.5 km for $\Delta T_1$ = 1 h and 28 km for $\Delta T_2$ = 3 h. If one calculates the divergence using only the position change

after 24 hours, the required base is 2.95/$a$ h$^{-1}$, and for $\sigma_{div}$ = 0.00085 h$^{-1}$ one obtains $a$ =3.5 km. Hence, by choosing a larger time interval, acceptable uncertainties can be obtained over smaller spatial scales. If positions acquired at shorter time intervals are available, they can be used for controlling the temporal evolution of the ice drift. Using $\sigma_{coord}$ = 2 m instead of 25 m in the example given above, we obtain $\sigma_{div}$ = 5.7/$a$ h$^{-1}$ for $\Delta T$ = 1 h, i.e. a base length of 6.7 km for a single measurement with $\sigma_{div}$ = 0.00085 h$^{-1}$, and 2.2 km for one measurement with $\Delta T$ = 3 h. Itkin et al. (2017) used triangle arrays with the smallest distance

between two buoys of 2 km, and the largest of 70 km.

Since the area and shape of the triangle change under the action of continuous stress, the uncertainty does not simply decrease by a factor of $1/\sqrt{n}$, i.e. with the number $n$ of buoy position readings. If we assume that the three-buoy array keeps the shape of an equal-sided triangle for 24 hours, with an increase in side length from $a_0$ to $1.1a_0$ (i.e. the area of the triangle increases by a factor of 1.05), the uncertainty of the last single measurement at the end of the 24 hour period is lower by a

factor of 1/1.1=0.91, Eq. (27, third term). Here it is assumed that the divergence is constant, the ratio $u_x/v_y$ is fixed by the ratio between base $a$ and height $h_a$ of the triangle, and the vorticity is zero, i.e. $u_y=v_x=0$. As mentioned above, the position error may be as small as 2 m.

### 3.6 Combination of grid cells or buoys

The combination of grid cells or several buoys is one possibility to lower the uncertainty of the area $\sigma_A$. In general,

the uncertainty of deformation rates is reduced when they are evaluated over a larger area, as can be deduced from the equations provided in Sects. 2.4 and 2.5. However, the uncertainty of the area, $\sigma_A$, appears explicitly only in the equations derived for the general case, Sect. 2.5. In Sects. 3.4 and 3.5 we showed that the terms including $\sigma_A$ can be neglected since velocity gradients observed for sea ice are usually small. Since, on the other hand, the change of the area inside a buoy array or of a grid cell can

also be used to quantify deformation (Lindsay and Stern, 2003), it is worthwhile to have a closer look at the effect of combining
several grid cells or buoys.

Because buoy arrays rarely reveal simple shapes such as squares or right triangles, the uncertainties in area have to be calculated numerically using Eq. (11) or (12). Hutchings et al. (2012), e.g., used 22 buoys, which they split into arrays of approximately equilateral triangles, but also into arrays of six, nine, and twelve buoys. Here we discuss combinations of squares and triangles.

First we investigate the effect of splitting a square or a right triangle into smaller units. We start with a square window covering $N \times N$ cells, i.e. we have $4N$ displacement vectors around it. Let $L'$ be the length of each side of a square covering several grid cells (Fig. 5). We divide each side of the square into $N$ segments of equal length. If $N = 2$ then each side of the square is 2 segments of length $L'/2$, and correspondingly for $N > 2$ it is $L'/N$. The term $\Sigma(x_{i+1} - x_{i-1})^2$ is zero if both $x_{i+1}$ and $x_{i-1}$ are located on the vertical sides of the square. On the top and bottom sides parallel to the $x$-axis, $N-1$ terms in the summation contribute $(2L'/N)^2$ for each side (indicated by green bars in Fig. 3). In addition, each corner contributes $(L'/N)^2$ (blue bars). The total contribution is hence $2(N-1)(2L'/N)^2 + 4(L'/N)^2 = 4(2N-1)(L'/N)^2$. The term $\Sigma (y_{i+1} - y_{i-1})^2$ contributes the same amount. Hence application of Eq. (12) yields:

$$\sigma_A^2 = \sigma_{coord}^2 (4N - 2) (L'/N)^2 \tag{28a}$$

Since each side of the square is divided into $N$ segments, the total number of points defining the boundary is $n = 4N$. With $L = L'/N$, we can rewrite Eq. (28a) as $\sigma_A^2 = \sigma_{coord}^2(n-2)L^2$. However, the notation in Eq. (28a), using $N$ and $L'$ instead of $n$ and $L$, is preferable because it explicitly shows that $\sigma_A^2$ decreases as $N$ increases for a fixed $L'$. Note that Eq. 28a is valid for buoy arrays. In case of SAR images, the tracking error has to be considered as well. When $\sigma_A^2$ is estimated for an area that deforms between acquisitions of SAR image 1 and 2, and the variance of the position error $\sigma_{coord}^2$ can be set to zero (see section 2.1), $\sigma_A^2 = 0$ for image 1. In image 2, however, it is $\sigma_A^2 = \sigma_{tr}^2(n-2)L^2$ (Lindsay and Stern, 2003).

For a right triangle, we have only two contributions from the corners instead of four as for the square (Fig. 5). In $x$-direction, e. g., the term $x_{i+4} - x_{i+2}$ is zero. Hence the total contribution in $x$- and $y$-direction is $4(N-1)(2L'/N)^2 + 4(L'/N)^2$ or

$$\sigma_A^2 = \sigma_{coord}^2 (4N - 3) (L'/N)^2 \tag{28b}$$

This can be written as $\sigma_A^2 / (\sigma_{coord}^2 L'^2) = (4N - 3) / N^2$, which takes the values 1, 5/4, 1 for N = 1, 2, 3, and then decreases as N increases. Note that the uncertainty initially increases from N=1 to N=2, and an improvement over N=1 is not reached until N=4.

In SAR applications, the question is whether it is preferable to use, e.g., the smallest possible ("elementary") square cell (determined by the resolution of the ice drift field) with four drift vectors at the edges, or to combine adjacent cells. Formally, the uncertainty in area for the elementary cell is $2\sigma_{coord}^2 L^2$, and for a cell with side length $L' = N \times L$, covering $N \times 4$ drift vectors, Eq. (28a) yields $\sigma_A^2 = \sigma_{coord}^2(4N-2) L^2$. Hence the uncertainty of the area increases when elementary cells are combined. However, since also the cell area increases by a factor of $N^2$, the single terms in Eqs. (13) – (22) that include the factor $A^{-2}$ decrease. When all terms except the fourth in Eqs. (23) and (24) can be neglected, it is immediately clear that the uncertainties of divergence and vorticity decrease if several elementary cells are combined into a larger square. The effect of local variations of the drift field on the deformation rate, however, can be considered in more detail when elementary cells (or smaller units of buoys arrays) are used for the calculations.

For buoy arrays it may be of advantage to use a larger number of buoys along the outline of a polygon. Here we study the example of an isosceles triangle with two sides of equal length (Fig. 6), which, e.g., comes closest to the array / subarrays

used by Hutchings et al. (2012). The term $\Sigma(x_{i+1} - x_{i-1})^2$ of Eq. (12) results in $(6N\text{-}4.5)(L'/N)^2$, for the term $\Sigma(y_{i+1} - y_{i-1})^2$ we obtain $(8N\text{-}6)(h/N)^2$. The areal uncertainty is hence:

$$\sigma_A^2 = (\sigma_{coord}^2/2)\,[(3N - 2.25)\,(L'/\text{N})^2 + (4\text{N-}3)(h/\text{N})^2] \tag{29}$$


Compared to an array consisting of three buoys at the edges of the triangle, the uncertainty can be reduced for $N \geq 4$, i.e. at least 12 buoys are required along the outline of the triangle. This also applies to the use of SAR images, when drift fields are retrieved from triangular cells.

If the shape of an array with many buoys approximately approaches the shape of a circle with radius $r$, and if the sum

of two line segments $s$ connecting vertices with summation index $i+1$ and $i$ differs only slightly compared to the chord length $s_c$ between vertices $i+1$ to $i\text{-}1$, the uncertainty of the area can be estimated as follows. We require that $s_c^2 \approx (2s)^2$. According to Eq. (12) the uncertainty in area is

$$\sigma_A^2 = \frac{\sigma_{coord}^2}{4}\sum_{i=1}^{n} s_{ci}^2 = \frac{\sigma_{coord}^2}{4}ns_c^2 \approx \frac{\sigma_{coord}^2}{4}n4s^2 \approx \sigma_{coord}^2\,n\left(r\,\frac{2\pi}{n}\right)^2 = \frac{4\sigma_{coord}^2}{n}\pi^2 r^2 \tag{30}$$


in agreement with Jansson and Persson (2014, Eq. (29)). Here we use the relationship $s = 2\pi r/n$ and take into account that both the even chords ($i\text{-}1$, $i+1$) with $i$=1,3,5,... and the odd chords with $i$=2,4,6,... each approximate the total perimeter of the circle (see e.g. Fig. 2, hexagon). To calculate the number of chords that is required to fulfill Eq. (30), we demand that $n's_c(1+e) = 2\pi r$, with $n'= n/2$, and $e$ is the error between the perimeter of a regular polygon and a circle. With $s_c/r = 2\sin(\pi/n')$ the condition

is $\sin(\pi/n')(1+e) = \pi/n'$. If $n'$=10 (i.e. a circled-shaped array with 20 buoys), $e$ is $< 0.017$.

## 3.7 Validity

It has to be kept in mind that the fundamental Eqs. (1), (2), (4) and (5) that we used for estimating the statistical uncertainties in the retrieval of deformation parameters are based on simplifying assumptions. Hence it is necessary to consider their range of validity when applying them.

### 3.7.1 Truncation error

The right-hand side of Eq. (5) for estimating $u_x$ is based on the trapezoid rule applied to the contour integral on the left side. The trapezoid rule is exact if $u$ is linear in $x$ and $y$; otherwise, the non-linear part of $u$ gives rise to a truncation error. Define segment $k$ of the contour integral to be the straight line from $(x_k, y_k)$ to $(x_{k+1}, y_{k+1})$, and define $\Delta x_k = x_{k+1} - x_k$ and $\Delta y_k = y_{k+1} - y_k$. Then segment $k$ of the contour integral $\oint u\,dy$ is estimated by $\frac{1}{2}(u_{k+1} + u_k)\Delta y_k$, as in Eq. (5), and the associated error

is:

$$e_k = -\frac{1}{12}\left(u_{xx}\Delta x_k^2 + 2u_{xy}\Delta x_k\Delta y_k + u_{yy}\Delta y_k^2\right)\Delta y_k \tag{31}$$

where the partial derivatives are evaluated at some point on segment $k$ (Atkinson, 1989). As can be seen, if $u$ is linear in $x$ and

$y$ on segment $k$ then $e_k = 0$. Similar error expressions apply to the estimates of the other velocity derivatives.

Higher-order estimates for $u_x$ could be derived, but they would not necessarily be more accurate because the ice motion may not be continuously differentiable to higher order, e.g. $u_{xxx}$ and higher derivatives may not exist. Higher-order estimates would only be more accurate for sufficiently differentiable fields.

### 3.7.2 Spatial resolution

Equation (5) provides an area-averaged estimate of $u_x$. The question arises as to whether the spatial resolution (i.e. the area) is small enough to capture the spatial variability in $u(x, y)$. One way to answer this question is to sub-divide the region into smaller pieces and repeat the calculation of $u_x$ for each piece. If the variability of $u_x$ from piece to piece is large then the sub-division of the original area was necessary; otherwise, it was not. In practice, sub-dividing a region means adding new data points, which is not always possible, unless the original region is purposely chosen to consist of the union of several

smaller pieces. An alternative method for determining whether the spatial resolution is adequate is given at the end of Sect. 3.8 below.

### 3.7.3 Temporal sampling

What temporal sampling is necessary to resolve changes in the sea-ice velocity field? The velocity may be decomposed into a mean field and a fluctuating part (Thorndike, 1986). Rampal et al. (2009) showed that the variance of the

fluctuating part has two regimes separated by a time scale of ~1.5 days. Since buoys deployed on sea ice report their positions every few hours or less, their sampling frequency is sufficient to resolve the velocity and its fluctuations. The revisit time of modern satellite constellations such as Sentinel-1 is less than a day at the high latitudes of the poles but older systems with three-day sampling may have missed some of the deformation caused by spatial variations in those fluctuations.

### 3.7.4 Correlation of errors

We have assumed that different error sources are uncorrelated and hence we have ignored the second term on the right-hand side of Eq. (2). While it is often true that spatial errors are uncorrelated with temporal errors, it may not always be the case that spatial errors are uncorrelated with each other. For example, for the distance $d = \sqrt[2]{(x' - x)^2 + (y' - y)^2}$ between two points $(x', y')$ and $(x, y)$, the full error variance of $d$ is given by:

$$\sigma_d^2 = \left(\frac{\partial d}{\partial x'}\right)^2 \sigma_{x'}^2 + \left(\frac{\partial d}{\partial x}\right)^2 \sigma_x^2 + \left(\frac{\partial d}{\partial y'}\right)^2 \sigma_{y'}^2 + \left(\frac{\partial d}{\partial y}\right)^2 \sigma_y^2 + 2\left(\frac{\partial d}{\partial x'}\right)\left(\frac{\partial d}{\partial x}\right)\sigma_{x'x}$$
$$+2\left(\frac{\partial d}{\partial y'}\right)\left(\frac{\partial d}{\partial y}\right)\sigma_{y'y} + 2\left(\frac{\partial d}{\partial x'}\right)\left(\frac{\partial d}{\partial y'}\right)\sigma_{x'y'} + 2\left(\frac{\partial d}{\partial x}\right)\left(\frac{\partial d}{\partial y}\right)\sigma_{xy} + 2\left(\frac{\partial d}{\partial x'}\right)\left(\frac{\partial d}{\partial y}\right)\sigma_{x'y} + 2\left(\frac{\partial d}{\partial x}\right)\left(\frac{\partial d}{\partial y'}\right)\sigma_{xy'} \quad (32)$$

If the coordinate uncertainties are all equal ($\sigma_x = \sigma_y = \sigma_{x'} = \sigma_{y'} = \sigma_{coord}$), and the covariances are all equal ($\sigma_{xy} = \sigma_{x'y'} = \sigma_{x'y} = \sigma_{xy'} = \sigma_{y'y} = \sigma_{x'x} = c$), then we obtain $\sigma_d^2 = 2\sigma_{coord}^2 - 2c$. Since the correlation between, e.g., $x$ and $y$ (and correspondingly for all

combinations above) is $\rho = \sigma_{xy}/(\sigma_x\sigma_y) = c / \sigma_{coord}^2$ we obtain $\sigma_d^2 = 2\sigma_{coord}^2(1 - \rho)$. In this case, a positive correlation serves to reduce $\sigma_d^2$ while a negative correlation serves to increase it. Since position errors are more likely to be positively correlated (due to systemic bias), ignoring the correlation terms is actually a conservative approach to error estimation.

### 3.7.5 Velocity discontinuities

When calculating uncertainties of deformation parameters, it is implicitly assumed that the sea-ice velocity does not

have discontinuities within the polygon in which the deformation is being estimated. This is because we use Eq. (5), which is based on Green's theorem. Numerous observations of the sea-ice velocity field show narrow shear zones or "linear kinematic features" (e.g. Kwok, 2003; Marsan et al., 2004; Kwok, 2006) across which the velocity jumps abruptly, as a result of stresses in the ice that create leads and ridges. Some researchers, e.g. Griebel and Dierking (2017) have proposed methods to detect and isolate these discontinuities in the velocity field to avoid smoothing effects when averaging adjacent velocity vectors (e.g.

for replacing outliers).

When applying Eq. (5) over an area with a discontinuity in the velocity field, a step-like function occurring between two positions $r_{i+1}$ and $r_i$ with $r_i = (x_i, y_i)$ is instead represented by a linear gradient. As the interval $\Delta r$ is decreased, the gradient increases. Hence, there is a numerical scaling effect: e.g. divergence and shear increase when calculated on grids of velocity vectors with higher spatial resolution. A discontinuity can be defined by a threshold for the difference of the velocities on both sides of it. The threshold depends on realistic values of velocity gradients in sea ice, and on the spatial resolution of the grid. The detection of possible discontinuities in a discrete field of velocity vectors, e.g. retrieved from SAR images, is helpful for the interpretation of the magnitudes of deformation.

## 3.8 Alternative method of analysis

In Sect. 2, the area-averaged velocity derivatives in a region are obtained by estimating contour integrals of the velocity around the boundary of the region. Two alternatives to this boundary integral ("BI") method are briefly discussed here: the least squares ("LS") method and the finite difference ("FD") method.

In the LS method, the velocity components $u$ and $v$ are modeled as linear functions of $x$ and $y$, plus error. Suppose velocities $(u_k, v_k)$ are given at locations $(x_k, y_k)$ for $k = 1$ to $n$. The linear model is:

$$u_k = A + B\,x_k + C\,y_k + \varepsilon_k \tag{33a}$$

$$v_k = D + E\,x_k + F\,y_k + \delta_k \tag{33b}$$

where the constants $A$, $B$, $C$, $D$, $E$, $F$ are chosen to minimize the variance of the errors $\varepsilon_k$ and $\delta_k$. The velocity derivatives $u_x$ and $u_y$ are then $B$ and $C$, while $v_x$ and $v_y$ are $E$ and $F$. The next step is to check whether the linear model accounts for a reasonable fraction of the variance in $u_k$ and $v_k$ by computing the squared correlation, and then whether the linear model does in fact provide a good fit to the data (by examining the spatial pattern of the errors $\varepsilon_k$ and $\delta_k$), or whether a quadratic or other non-linear model is more appropriate.

The FD method provides an estimate of $u_x$ (and the other velocity derivatives) at a single point, based on Taylor series expansions of $u$ and $v$ about that point. For example, suppose we have velocities $u_{k+1}$ and $u_{k-1}$ at locations $(x_{k+1}, y)$ and $(x_{k-1}, y)$, where $x_k = x_0 + k\Delta x$. Then an estimate of $u_x$ at $(x_k, y)$ is:

$$u_x^{FD} = \frac{u_{k+1} - u_{k-1}}{2\Delta x} = \frac{u(x_k + \Delta x, y) - u(x_k - \Delta x, y)}{2\Delta x} = u_x + \frac{1}{6} u_{xxx} \Delta x^2 + O(\Delta x^4) \tag{34}$$

where the derivatives on the right-hand side are evaluated at $(x_k, y)$. The first term on the right-hand side is the true value of $u_x$ at $(x_k, y)$; the rest of the terms are the truncation error, i.e. error $= u_x^{FD} - u_x = (1/6)\, u_{xxx}\, \Delta x^2 +$ higher-order terms.

In summary, the BI method provides area-averaged estimates of $u_x$, $u_y$, $v_x$, $v_y$; the LS method provides the best linear models of $u$ and $v$, from which $u_x$, $u_y$, $v_x$, $v_y$ follow; and the FD method provides point estimates of $u_x$, $u_y$, $v_x$, $v_y$.

For a rectangular region with velocities given only at the four corners, it turns out that all three methods give the same estimates of $u_x$, $u_y$, $v_x$, $v_y$, assuming the FD estimate is made at the center of the rectangle. For a general configuration of points, the three methods give different estimates. Note that in the BI method, velocities inside the boundary of the region are ignored. In the LS method, velocities farther from the mean location $(\bar{x}, \bar{y})$ have greater weight in determining the slope of the linear model. The FD method is most appropriate for regularly-spaced square grids, whereas the BI and LS methods are equally applicable to irregular grids.

The LS method can be used as a diagnostic tool to determine whether the spatial resolution of the velocity data adequately captures the variability of the velocity field. Analysis of the spatial pattern of the LS residuals (errors) by standard methods (autocorrelation) reveals whether the linear velocity model is in fact a good fit to the velocity data or not. If it is a

good fit, then the spatial resolution is adequate, and the truncation error in the BI method is small. If it is not a good fit and sufficient data are available, the region should be divided into smaller pieces and the calculation repeated for each piece. The BI method should be used to calculate the actual (area-averaged) velocity derivatives, since it does not depend on a model that

needs to be checked for goodness of fit.

## 4. Conclusions

In this study we derived equations for calculating the magnitude of different deformation parameters within a given area, using displacement vectors retrieved from SAR images or buoy arrays. In the most general case, presented in Sect. 2.5, errors in measurements of position ("geolocation error"), velocity (determined from displacement), and area size have to be

considered. Uncertainties in velocity and area size can be related to uncertainties in position measurements and (for velocity) time readings (Sects. 2.2 and 2.3). When retrieving displacements from pairs of SAR images a tracking error has to be considered additionally.

In Sect. 3, uncertainties of divergence and vorticity are derived based on the general equations introduced in Sect. 2, assuming squares and triangles as outlines for the area over which deformation is calculated. We chose these geometric shapes

since they have been frequently used in past and recent studies of deformation in sea ice. The major findings are as follows.

- The equations reveal that the uncertainties in divergence and vorticity increase with the magnitudes of the velocity gradients, and with the geolocation and tracking errors. They decrease with increasing size of the area and the time interval $\Delta T$ used for calculating the velocity gradients (Sects. 3.4 and 3.5). These results agree with the recent work of Bouchat and Tremblay (2020). Since uncertainties of shear and total deformation are weighted averages of divergence and vorticity

(Sect. 2.5), the conclusions drawn for the latter are also valid for the former.

- Since geolocation errors in SAR images are usually correlated over scales of $\geq 10$ km they can be treated as a constant bias. In this case, position uncertainties are relatively small and may even be set to zero (Sects. 2.1, 2.4, 3.7.4).

- Geolocation errors in imaging modes of modern SAR systems are smaller than their spatial resolution (see Sect. 3.4.1). Errors in time readings of buoy positions and SAR image acquisitions are negligible in most cases. For buoy arrays, the

magnitude of the position error may not be negligible. Here, the reader is advised to check the manual for the position sensor and pay attention to whether the error is given as standard deviation or in another format.

- The tracking error that needs to be considered for displacement fields retrieved from SAR images is on the order of the length of one pixel, as several studies showed. If the geolocation error can be neglected relative to the tracking error, a good approximation for the uncertainty of divergence and vorticity valid for a square with side $L$ or a triangle with base $L$

is $\sigma = a \times \sigma_{tr}/(\Delta T \times L)$, where $a = \sqrt{2}$ for the square and $a = 2$ for the triangle. If squares or triangles are small, the ratio $\sigma_{tr}/L$ and hence the uncertainty is large.

- For a given threshold of acceptable uncertainty we estimated the necessary size of rectangular grid cells in SAR images and triangular buoy arrays, focusing on divergence and vorticity as examples (Sects. 3.4.2 and 3.5.2.). At larger temporal sampling rates, the areas can be made smaller.

- The area uncertainty of the smallest possible ("elementary") cell, determined by the position of three or four adjacent displacement vectors at the edges of a triangle or square, is smaller than for a group of adjacent elementary cells with more displacement vectors on the perimeter around the group (Sect. 3.6). If, on the other hand, for an area of fixed size a variable number $N$ of displacement vectors can be selected, the area uncertainty normally decreases with increasing N. For triangles, however, we found that the area uncertainty with 6 displacement vectors is larger than the one with three

(see Sect. 3.6 for details).

- In Sects. 3.7 and 3.8 we provided thoughts concerning the validity of the derived equations, which assume that the velocity field inside elementary cells is continuous and can be approximated by a two-dimensional linear function. By including

second-order terms or carrying out least-square fits over sub-regions of the velocity field, the validity of linearity can be judged. In the former case the second-order terms need to remain below a certain threshold, in the latter, the correlation coefficient should be large. Discontinuities in the velocity field should be detected before deformation is calculated to allow their impact to be assessed and to consider appropriate strategies to alleviate their impact.

**Acknowledgements**. The work of WD was partly funded by CIRFA (Center for Integrated Remote Sensing and Forecasting for Arctic Operations): RCN research grant no. 237906. HLS was funded by NASA grant NNX17AD27G, and JKH by National Science Foundation grant NSF 1722729. We thank one anonymous reviewer and Amélie Bouchat for comments that led to improvements in the manuscript.

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

Code/Data availability: does not apply

Author contribution: WD and HS derived equations and discussed the validity of the approach, WD and JH collected and evaluated typical ranges of measurement parameters, all authors developed the concept of the study and worked on the text

Competing interests: none


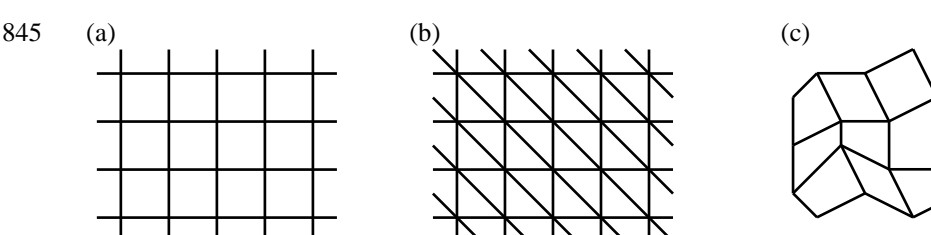


**Figure 1:** Eulerian grids (a) and (b) are re-initialized at every time step to a regular configuration. Lagrangian grids (c) evolve over time without being re-initialized.


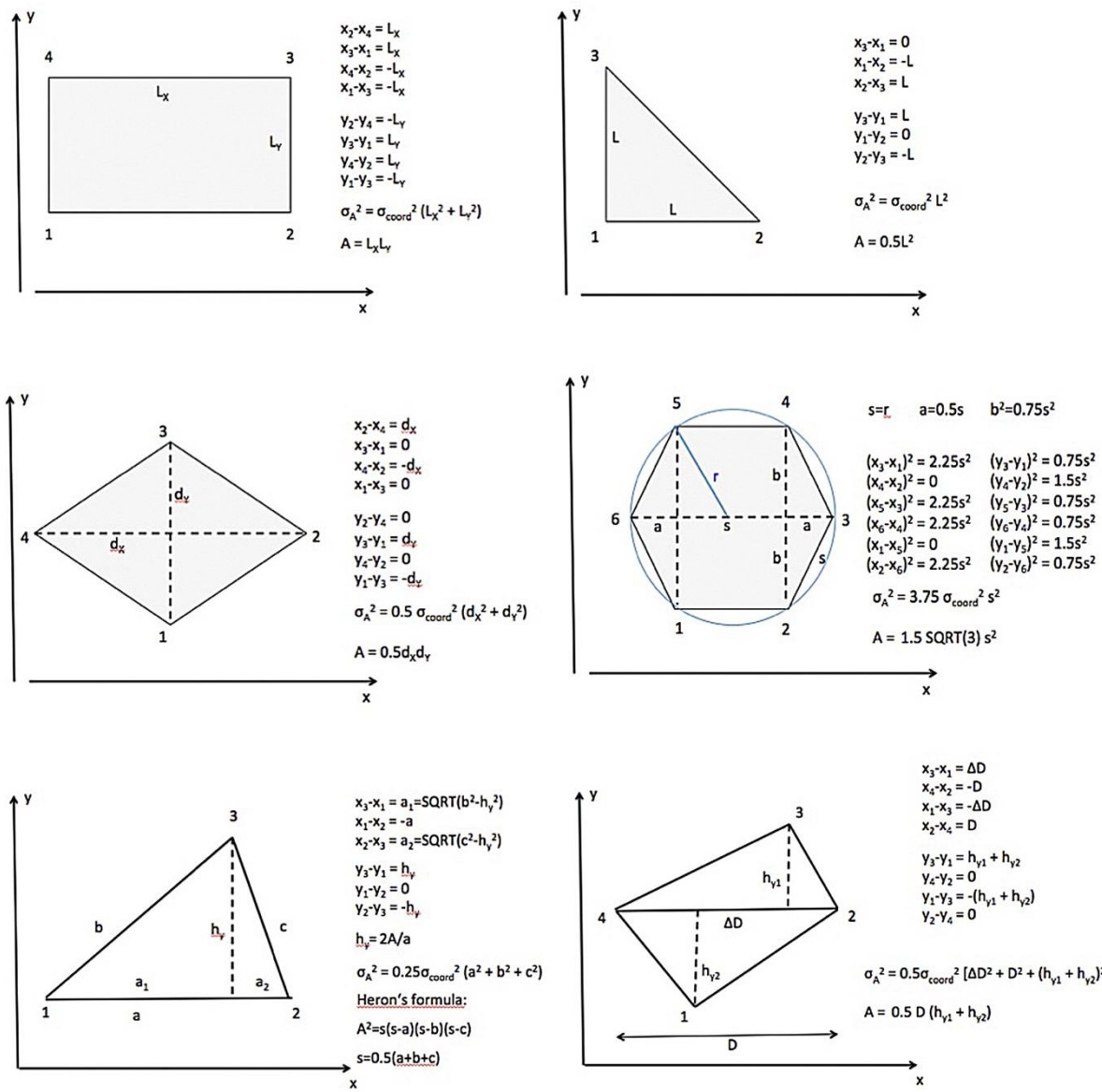

**Figure 2:** Application of Eq. (12) to different geometrical figures: rectangle, equal-sided right triangle, rhombus, regular hexagon, triangle, and quadrangle.


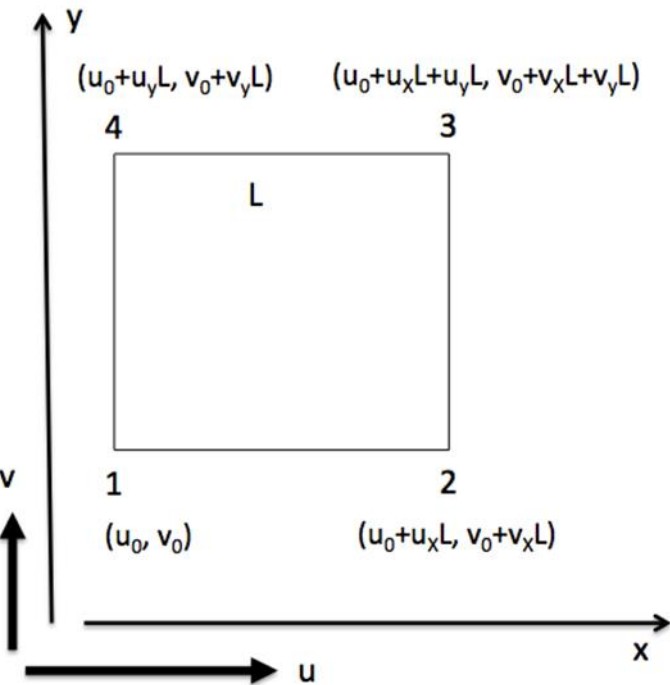

**Figure 3.** Uncertainty of divergence and vorticity for a square in a spatially varying velocity field with gradients $u_x$, $u_y$, $v_x$, $v_y$.



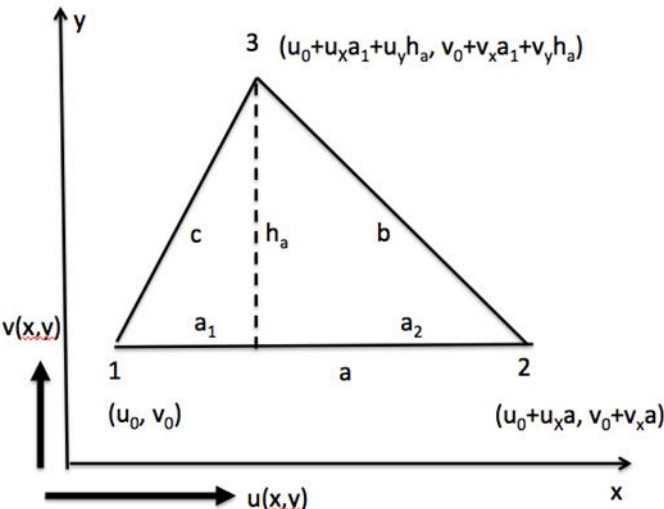

**Figure 4.** Uncertainty of divergence for a triangle in a spatially varying velocity field with gradients $u_x$, $u_y$, $v_x$, $v_y$. The height $h_a$ is $2A/a$ ($A$ can be calculated from Heron's formula), and $a_1 = c^2 - h_a^2$. Side $a$ is the base of the triangle.

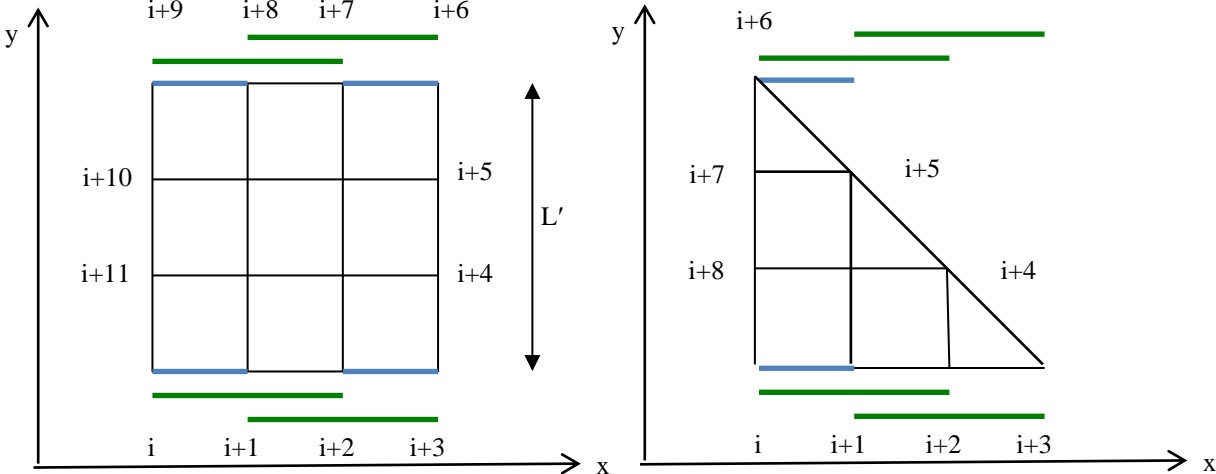


**Figure 5:** Derivation of equation 30 in *x*-direction for *N* = 3. Green and blue bars indicate terms to be considered in the derivation of Eqs. (28a) and (28b).

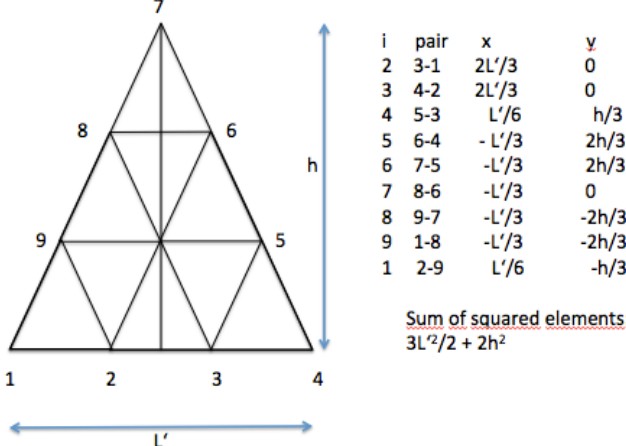

| i | pair | x | y |
|---|------|-----|------|
| 2 | 3-1 | 2L'/3 | 0 |
| 3 | 4-2 | 2L'/3 | 0 |
| 4 | 5-3 | L'/6 | h/3 |
| 5 | 6-4 | - L'/3 | 2h/3 |
| 6 | 7-5 | -L'/3 | 2h/3 |
| 7 | 8-6 | -L'/3 | 0 |
| 8 | 9-7 | -L'/3 | -2h/3 |
| 9 | 1-8 | -L'/3 | -2h/3 |
| 1 | 2-9 | L'/6 | -h/3 |

Sum of squared elements:
$3L'^2/2 + 2h^2$

**Figure 6.** Application of Eq. (12) on a triangle with two equal sides for $N = 3$.

Table 1: Magnitudes of terms 1 to 4 in Eqs. (23) and (24)

| Reference | Hollands and Dierking (2011) | Hollands and Dierking (2011) | Lindsay and Stern (2003) | Lindsay and Stern (2003) |
|---|---|---|---|---|
| Image mode | ASAR IM $\sigma_{coord}$ = 1 m L=300 m | ASAR WS $\sigma_{coord}$ = 7 m L=1200 m | Radarsat ScanSAR, assump. $\sigma_{coord}$ = 0 L=10 km | Radarsat ScanSAR $\sigma_{coord}$ = 190 m L=10 km |
| 1. term | $< 3.33 \times 10^{-5}$ d$^{-2}$ | $< 1.02 \times 10^{-4}$ d$^{-2}$ | 0 | $< 1.2 \times 10^{-3}$ d$^{-2}$ |
| 2. term | $< 1.11 \times 10^{-5}$ d$^{-2}$ | $< 3.40 \times 10^{-5}$ d$^{-2}$ | 0 | $< 4.0 \times 10^{-4}$ d$^{-2}$ |
| 3. term<br>$\Delta T$ = 1 d<br>$\Delta T$ = 3 d<br>$\Delta T$ = 6 d | $4.44 \times 10^{-5}$ d$^{-2}$<br>$0.49 \times 10^{-5}$ d$^{-2}$<br>$0.12 \times 10^{-5}$ d$^{-2}$ | $1.36 \times 10^{-4}$ d$^{-2}$<br>$1.51 \times 10^{-5}$ d$^{-2}$<br>$0.38 \times 10^{-5}$ d$^{-2}$ | 0 | $1.78 \times 10^{-4}$ d$^{-2}$ |
| 4. term<br>$\Delta T$ = 3 d, $\sigma_{tr}$ = 100 m<br>max: $\Delta T$ = 1 d, $\sigma_{tr}$ = 40 m<br>min: $\Delta T$ = 6 d, $\sigma_{tr}$ = 20 m<br>max: $\Delta T$ = 1 d, $\sigma_{tr}$ = 240 m<br>min: $\Delta T$ = 6 d, $\sigma_{tr}$ = 120 m | $3.56 \times 10^{-2}$ d$^{-2}$<br>$2.47 \times 10^{-4}$ d$^{-2}$ | 0.08 d$^{-2}$<br>$5.56 \times 10^{-4}$ d$^{-2}$ | $2.2 \times 10^{-5}$ d$^{-2}$ | $2.2 \times 10^{-5}$ d$^{-2}$ |