# Peer review of "Estimating statistical errors in retrievals of ice velocity and deformation parameters from satellite images and buoy arrays"

_The Cryosphere, 2020_

## Referee Comment (RC1) · Anonymous Referee #1 · 5 Mar 2020

This paper presents a consistent framework for estimating statistical errors of sea ice drift and ice deformation parameters derived from a set of GPS on-ice buoys or from sequential satellite SAR images. Throughout the case studies, the authors carefully examined various sources of errors and their estimates relevant to the both main types of sea ice drift observations. This paper will serve as a good reference for the future studies dealing with deriving kinematic parameters of sea ice and the corresponding statistical errors. The paper is well structured and written. I recommend it to publication, once the following comments are addressed:

In Section 3.4 the authors discuss deformation parameters retrieval for square grid

cells in SAR images. However, many of the recent algorithms for retrieval of ice drift information from SAR images (e.g., Demchev et. al., 2017; Muckenhuber et al., 2016; Komarov and Barber, 2014) compute ice motion on a non-square grid, as usually the grid points are associated with distinctive ice features in SAR images. Could the authors extend their analysis (Section 3.4) from the "square" grid cells to "non-square" grid cells in SAR images? What is the most accurate approach to computing ice deformation parameters and associated errors from the SAR-derived ice velocities provided on non-square grid cells?

Technical corrections:

Line 39-40. I suggest modifying:

"This means that drift and deformation errors do not only depend on the geolocation accuracy of satellite images but also on the reliability and robustness of the drift retrieval algorithm."

to

"This means that drift and deformation errors do not only depend on the geolocation accuracy and spatial resolution of satellite images but also on the reliability and robustness of the drift retrieval algorithm."

Line 76-77. "x" and "y" should not be in bold as they are scalars.

Line 141. "10km" –> "10 km"

Line 457-467. Same as previous. Add spaces between numbers and units.

Equation (31). N should not be in bold.

Line 725. "RADARSAR" –> "RADARSAT"

---

## Referee Comment (RC2) · Amelie Bouchat (Referee) · 8 Apr 2020

**Review of "*Estimating statistical errors in retrievals of ice velocity and deformation parameters from satellite images and buoy arrays*", Dierking et al.**

**By Amélie Bouchat**

The manuscript presents a general mathematical background for estimating the error of sea-ice drift velocities and deformation rates obtained from buoys or synthetic aperture radar (SAR) satellite imagery. While a majority of derivations were already presented in previous studies, the authors gather and discuss here all aspects of the error analysis in a single study and also provide examples for specific cases that can help the reader understand how to apply the derivations for a variety of scenarios. Additionally, the authors present alternative methods (other than the usual boundary integral definition) for estimating the sea-ice deformation rates from observations. Given that the observed deformation fields from SAR or buoy products are recently being used in the community to evaluate sea-ice models and/or infer material properties of geophysical sea-ice, providing a complete framework for quantifying the uncertainty on the drift and deformation fields is timely and highly valuable. The paper is generally well-written and I recommend it for publication in *The Cryosphere*, after the following comments are addressed in the revisions.

**General comments:**

A lot of equations derived in Section 2 have already been derived in previous publications, but references are not always provided (e.g. Eq. (12) is already in Lindsay and Stern, 2003; and Eq. (13) is also in Griebel and Dierking, 2018;). I would also like to note that Bruno Tremblay and I have recently submitted a paper (December 2019, currently under review at the Journal of Geophysical Research: Oceans) in which we also present and discuss the same equations as the Equations (19)-(24) in Section 2.5 in the present paper, but applied to the RADARSAT Geophysical Processor System Lagrangian drift data set only. Both H. Stern and J. Hutchings are aware of this work since they have been involved in discussions or review of this work. As mentioned to me already by H. Stern, we should cite each other's work in our revisions (I will send you a copy of the pre-print once it is accepted).

Different cases and examples for specific observation products are also discussed in the manuscript, which is useful but it is hard to extract the conclusions/main points from these examples in the text. I would therefore suggest the addition of tables/figures to convey these conclusions more clearly. For example, a table presenting the geolocation, tracking, and timing errors for the different SAR and buoy products as mentioned in Sections 3.4 and 3.5 would be a very useful reference for the reader and for future studies. Below, I also suggest presenting the dependence of the error on the number of tracked points in a graphical form so that it can be used to guide future studies on choosing how many points should be considered.

**Specific Comments:**

P1. Line 36: *"The accuracy of deformation parameters.."*
Sea-ice deformation should be defined in the introduction (i.e. shear, divergence, etc.) before mentioning their errors. It is also not clear what "deformation parameters" are. Do you refer to the shear, convergence, divergence, etc.? In this case, I would change simply to "deformations" or "deformation rates" for consistency with previous studies.

P.1 Lines 35-37: *"For buoys, errors in drift measurements depend on [...] but also by the size and shape of buoy arrays."*
References should be added here, e.g. Hutchings et al. (2012), Griebel and Dierking (2018).

P1. Line 41: *"The issue of error estimation was repeatedly addressed in the past, scattered in a number of publications [...]"*
Also add Bouchat and Tremblay (2020, under review).

P.2 Line 50: *"for calculating errors of drift and deformation parameters, supplemented with the derivation of general-case uncertainties of divergence, vorticity, shear, and total deformation."*
Again, it is not clear what is the difference between "deformation parameters" and "divergence, vorticity, shear, and total deformation".

P.4 Lines 139-142: *"If, e.g., the distance between two moving objects is closer than this, the position errors cancel and $\sigma_d^2 = \sigma_{tr}^2$ for the retrieval from a SAR image pair and $\sigma_{coord}^2 = 0$ between two buoys. Hence within a circle of 10km or less in diameter, deformation can be estimated with sufficient accuracy even if geolocation errors are high."*
I don't understand how the geolocation errors cancel given that they are squared and add up when using the propagation of error on $d$ to obtain $\sigma_d^2 = 2\sigma_{coord}^2$. Can you explain?

P4. Line 157: *"considering that U=..."*
I think you mean $U = d/\Delta T$? Otherwise, you would have to use Eq. (7) to get $\sigma_U^2$.

P.6 Line 225: *"Throughout this section we assume that $\sigma_U = \sigma_u = \sigma_v$."*
because you assume $\sigma_{\Delta T} = 0$? If so, please mention it.

P.7 Eq. (18): The term $\sigma_{coord}$ should not appear in this equation since it was assumed that it is equal to zero since the beginning of this section. If not zero, then other terms should appear in Eqs. (13) & (14) and Eq. (18) to account for the error on the area and position in the strain rate definition explicitly.

P.8 Equations (23) and (24): These are the same as Eq. (15) and (16) presented earlier. Remove and refer to Eq. (15)-(16) instead?

P.8 Line 295: "*Hence, the uncertainties of divergence, vorticity, shear, and total deformation differ from one another.*"
Unless $\sigma_{u_x} \sim \sigma_{u_y} \sim \sigma_{v_x} \sim \sigma_{v_y}$ , then they are equal.

P.8 Line 305: The section is titled "*Typical uncertainties of deformation parameters*" but the section does not describe uncertainties but rather a short review of observed deformation rates from previous studies. If the purpose of this section is to describe observations of shear, divergence, etc. then it should be retitled, and it should also discuss the fact that the observed deformation magnitudes are closely tied with the scale of observation given that the mean deformation rate is known to decay following a power-law with increasing spatial and temporal scale (e.g. Marsan et al., 2004; Rampal et al., 2008; Stern and Lindsay, 2009; Bouchat and Tremblay, 2020 - see at the end of this document for the references if not already in your list).

P.9 Line 322: "*The first term in Eqs. (21) and (22) is smallest if, for a given area, $\sigma_A$ is at a minimum.*"
and for a given value of $u_x^2 + v_y^2$, or if $u_x^2 + v_y^2$ is also at a minimum.

P.9 Section 3.2: It would be interesting to show a graph of the ratio $\sigma_A^2/\sigma_{coord}^2$ as a function of the number of tracked points for fixed values of *A (e.g. 1 km², 10² km², 20² km², 100² km², etc.)* to complement the discussion. It seems like going from three points to four points (i.e. from triangle to square) increases the error contribution of this term, but then going from 4 points to 6 points (i.e. square to hexagon) reduces the contribution of this term to the global error. It could be useful to see this in a functional form to guide choosing (if possible) a reasonable number of tracked points to reduce the area error. This could be added in Section 3.6.

P.9 Line 340: "*For a given position error,...*"
and a given tracking error

P.9 Line 341: "*The third term is solely dependent on the coordinate uncertainty $\sigma_{coord}$.*"
No, the third (last) term also decreases with increasing area *A*.

P.10 Line 361: "*When ice drift is retrieved from SAR images, the contribution of those terms that depend on $\sigma_{coord}/L$ can usually be neglected.*"
Lindsay and Stern (2003) report that previous estimates of the geolocation error for the RADARSAT ScanSAR images are of the order of ~200 m, hence non-negligible when compared to the tracking error (~100 m). So it is not negligible for all SAR products. It would be worth including those estimates for RADARSAT as well since this product is often used to obtain the observed sea-ice deformation fields. In fact, in Bouchat and Tremblay (2020, under review), we show that when using all the other terms except the tracking error for the RGPS data set, the resulting error on the total deformation rates can be ~1.5 times larger than Eq. (17) in Lindsay and Stern, 2003 (or the equivalent tracking error term in your Eq. 25 and 26). In this case, the terms in $\sigma_{coord}/L$ cannot be neglected.

P.11 Line 395: "*with $\sigma_{tr}^2/2\sigma_{coord}^2$ approximately equal to $100^2/(2\times5^2) = 200$*"
Here it is assumed that $\sigma_{coord}$ = 5 m for RADARSAT ScanSAR images, but Lindsay and Stern (2003) mention a geolocation error that is on the order of ~ 200 m (see also comment above). Can you indicate where the value of $\sigma_{coord}$ = 5 m was taken from?

P.9-13 Sections 3.4 and 3.5: These two sections focus on estimation of errors for the divergence and vorticity, and compare the contribution of different terms to the total error on divergence and vorticity. However, shear and total deformation rates are often larger than divergence (see increased probability of larger deformation rates in PDFs of shear vs divergence in e.g. Bouchat and Tremblay 2017, or Stern et al. 1995). How does the interpretation of the importance of each term in the error formulation change when considering the error on shear and total deformation instead of divergence and vorticity? For this, it could also be useful to present the expanded version of Eq. (23) and (24).

I also found it hard/confusing to follow all the examples presented at the end of section 3.5. A lot of different cases and numbers are presented and it is easy to get lost in the conclusions that should be retained. A visual aid (such as a table or graph) that gathers the essential points that are supposed to be conveyed by these examples could be added for more clarity.

P.13 Line 489: "*Let L´ be the length of each side of the big square (Fig 5).*"
L' is not defined on Figure 5. And "*big square*" = "window" ?

P.13 Line 490: "*Because of the enclosed grid cells we can divide each side of the square window into N segments of equal length.*"
Is Eq. (30) derived here only valid when the grid cells are not moving and of equal length? For a Lagrangian grid, the cells are not necessarily of equal size and therefore Eq. (30) would not apply, and one would still need to evaluate the full expression in (12), correct?

P.13 Line 500: "*we can rewrite Eq. (30) as $\sigma_A^2 = \sigma_{coord}^2(n-2)L^2$ which is Eq. (16) in Lindsay and Stern (2003).*"
Equation (16) in Lindsay and Stern (2003) uses the tracking error $\sigma_{tr}^2$ instead of the geolocation error as considered here $\sigma_{coord}^2$ . It is confusing because in Section 3.4, it is mentioned (line 360) that the last term of Eq. (25) term depending on the tracking error is the same as Eq (17) in Lindsay and Stern (2003), but Eq. (17) in Lindsay and Stern is derived using their Eq. (16) which is now assumed to be using the geolocation error here… Can you please clarify? Also, please indicate if there is a mistake in Lindsay and Stern (2003).

P.13 Line 506: "*Note that $\sigma_A^2$ for the right triangle is $\sigma_{coord}^2 L'^2$ for N = 1, $1.25\sigma_{coord}^2 L'^2$ for N = 2, [...], and first with N = 4, the uncertainty can be reduced.*"
It could be worth presenting this discussion using a graphical form, for more clarity. See also previous comment for P.9 Section 3.2.

P.13 Line 513: "*Hence the uncertainty of the area increases when elementary cells are combined. However, since also the cell area increases by a factor of $N^2$, the single terms in Eqs. (13) – (22) that include the factor $A^{-2}$ decrease.*"
Which one wins? Is it better in the end to aggregate cells?

P.14 Line 517: "*For buoy arrays it may be of advantage to use a larger number of buoys along the outline of a polygon.*"
Couldn't SAR drift fields also be derived using triangle cells (in principle) and the discussion regarding Eq. (31) could therefore apply to both SAR and buoy applications?

P.14 Line 535: "*To calculate the number of chords that is required to fulfill Eq. (32), we demand that $n'_{sc}$ (1+e) = 2πr, with n'= n/2, and e the error.*"
This is unclear; "e" is the error on/of what?

P.14 Eq (33): It is not clear to me how you obtain this result. I can see that it probably involves a Taylor series expansion of u(x,y) however if I do this expansion around $(x_k, y_k)$, e.g.:

$$u(x,y) = u(x_k, y_k) + (x - x_k)u_x + (y - y_k)u_y + \frac{1}{2}\left[(x - x_k)^2 u_{xx} + 2(x - x_k)(y - y_k)u_{xy} + (y - y_k)^2 u_{yy}\right] + [...]$$

where the derivatives are evaluated $(x_k, y_k)$. Then I evaluate u(x,y) at $(x_{k+1}, y_{k+1})$ and use the same definitions of $\Delta x_k$ and $\Delta y_k$ as in the manuscript, and I get:

$$u_{k+1} = u_k + \Delta x_k u_x + \Delta y_k u_y + \frac{1}{2}\left[\Delta x_k^2 u_{xx} + 2\Delta x_k \Delta y_k u_{xy} + \Delta y_k^2 u_{yy}\right] + [...]$$

Such that, I get:

$$\frac{1}{2}\left(u_{k+1} + u_k\right)\Delta y_k = u_k \Delta y_k + \frac{1}{2}\Delta x_k \Delta y_k u_x + \frac{1}{2}\Delta y_k^2 u_y + \frac{\Delta y_k}{4}\left[\Delta x_k^2 u_{xx} + 2\Delta x_k \Delta y_k u_{xy} + \Delta y_k^2 u_{yy}\right] + [...]$$

So I see that my last term here is similar to your definition of $e_k$ but I don't know how to get there. Can you clarify?

P.16 Line 591: "*(see above)*"
Not clear to what this is referring to.

P.16 Line 624: "*For a general configuration of points, the three methods give different estimates.*"
Have you obtained numerical estimates for examples using each method? How much do they differ?

P.17 Section 4 Conclusions:
The first point has also been shown in Bouchat and Tremblay (2020, under review). The second point should also mention the exception for RGPS.

P. 19 Line 725: "*RADARSAR*" should be RADARSAT

**Formatting and writing suggestions:**

P.1 Line 16: "*in an array.*" → "in an array of buoys."

P.1 Line 19: "*also a tracking error has to be considered.*" → "a tracking error also has to be considered."

P.1, Line 24: "*the magnitudes of deformation parameters*" → "the magnitude of deformation parameters."?

P1. Line 33: "*sea ice mapping*" → "sea-ice mapping".

P. 2 Line 46: "*truncation error*" → change to "boundary-definition errors", or add it in parentheses to link with previously-used formulation? (or indicate why the previous formulation is incorrect).

P.4 Line 134: "*one needs to consider position and tracking uncertainties $\sigma_{coord}^2$ and $\sigma_{tr}^2$.*" → "one needs to consider position and tracking uncertainties, *i.e.* $\sigma_{coord}^2$ and $\sigma_{tr}^2$ *respectively.* "

P.6 Line 208: "*The cell covers m x m square-shaped pixels.*" → "The cell covers m x m square-shaped pixels of resolution $\Delta x$ ."

P.6 Lines 236-237: "*the sum of variances of the left term*" → "the sum of variances of the *first* term"? And "*the sum in the right term*" → "the sum in the *second* term"?

P.8 Line 286: "*For the shear one obtains...*" → "For the shear, one obtains..."

P8. Eq. (23): x and y in $\sigma_{u_x}$, $\sigma_{u_y}$, $\sigma_{v_x}$, and $\sigma_{v_y}$ should be subscripts.

P.9 Line 343: "*In the following discussion we assume that position data of all buoys are exactly synchronized but also discuss an example for which this was not the case.*"
Add reference to section 3.5 at the end of this sentence?

P.13 Line 485: "*the uncertainties have to be calculated numerically.*" → "the uncertainties have to be calculated numerically using Eq. (12)"?

P.14 Line 515: "*can be considered*" → "can *also* be considered"?

**Figures:**

Fig. (2): The figure is blurry.

Fig. (5): Please indicate in the label what are the blue and green lines.

**Additional References:**

Bouchat, A., and B. Tremblay (2020), Reassessing the quality of sea-ice deformation estimates derived from the RADARSAT Geophysical Processor System and its impact on the spatio-temporal scaling statistics, *Journal of Geophysical Research: Oceans* - under review

Bouchat, A., and B. Tremblay (2017), Using sea-ice deformation fields to constrain the mechanical strength parameters of geophysical sea ice, *Journal of Geophysical Research: Oceans,* doi:10.1002/2017JC013020.

Rampal, P., J. Weiss, D. Marsan, R. Lindsay, and H. Stern (2008), Scaling properties of sea ice deformation from buoy dispersion analysis, *Journal of Geophysical Research: Oceans*, 113(C3), doi:10.1029/2007JC004143.

Stern, H. L., and R. W. Lindsay (2009), Spatial scaling of arctic sea ice deformation, *Journal of Geophysical Research: Oceans*, 114(C10), doi:10.1029/2009JC005380, c10017.

Stern, H. L., Rothrock, D. A., and Kwok, R. (1995), Open water production in Arctic sea ice: Satellite measurements and model parameterizations, *Journal of Geophysical Research*, 100(C10), 20601– 20612, doi:10.1029/95JC02306.

---

## Author Comment (AC1) · 21 May 2020

This paper presents a consistent framework for estimating statistical errors of sea ice drift and ice deformation parameters derived from a set of GPS on-ice buoys or from sequential satellite SAR images. Throughout the case studies, the authors carefully examined various sources of errors and their estimates relevant to the both main types of sea ice drift observations. This paper will serve as a good reference for the future studies dealing with deriving kinematic parameters of sea ice and the corresponding statistical errors. The paper is well structured and written. I recommend it to publication, once the following comments are addressed:

Thank you. Our answers to your comments are marked in green.

In Section 3.4 the authors discuss deformation parameters retrieval for square grid cells in SAR images. However, many of the recent algorithms for retrieval of ice drift information from SAR images (e.g., Demchev et. al., 2017; Muckenhuber et al., 2016; Komarov and Barber, 2014) compute ice motion on a non-square grid, as usually the grid points are associated with distinctive ice features in SAR images. Could the authors extend their analysis (Section 3.4) from the "square" grid cells to "non-square" grid cells in SAR images? What is the most accurate approach to computing ice deformation parameters and associated errors from the SAR-derived ice velocities provided on non-square grid cells?

This is two separate questions.

1. Regarding non-square grid cells: see Section 2.5, in which uncertainties of deformation parameters are derived for a general polygonal region. These are based on the general formula for $u_x$ (and the other velocity derivatives) given in equation (5), which is based on integration around the boundary of an arbitrary region. In Section 3.4 we simply specialized the results of Section 2 to square grid cells. We also added a paragraph providing a more general view on the problem of arbitrarily-shaped quadrangles at the end of Section 3.4.

2. Regarding "the most accurate approach to computing ice deformation parameters and associated errors from the SAR-derived ice velocities provided on non-square grid cells," there are at least two factors to consider:
    (a) Tradeoff between accuracy and spatial resolution. We can apply equation (5) for $u_x$ (and the other velocity derivatives) over larger and larger regions defined by more and more boundary points to obtain more accurate estimates of the mean deformation, at the expense of reduced spatial resolution. In other words, one must balance the need for accuracy with the need for spatial resolution of the deformation field. This is mentioned in the second sentence of Section 3.6.
    (b) Differentiability of the velocity field. The truncation error of equation (5) is of order $u_{xx}\Delta x^2$, i.e. second-order accurate: it is exact for velocity fields that are linear in x and y. Higher-order estimates for $u_x$ could be derived, but they would not necessarily be more accurate because the ice motion may not be continuously differentiable to higher order, e.g. $u_{xxx}$ and higher derivatives may not exist. Higher-order estimates would only be more accurate for sufficiently differentiable fields. We have added the above sentences to the end of Section 3.7.1.

Technical corrections:

Line 39-40. I suggest modifying:
"This means that drift and deformation errors do not only depend on the geolocation accuracy of satellite images but also on the reliability and robustness of the drift retrieval algorithm."
to
"This means that drift and deformation errors do not only depend on the geolocation accuracy and spatial resolution of satellite images but also on the reliability and robustness of the drift retrieval algorithm."
Done.

Line 76-77. "x" and "y" should not be in bold as they are scalars.
Was perhaps a problem when generating the PDF from Word

Line 141. "10km" –> "10 km"
Done.

Line 457-467. Same as previous. Add spaces between numbers and units.
Done.

Equation (31). N should not be in bold.
Was perhaps a problem when generating the PDF from Word

Line 725. "RADARSAR" –> "RADARSAT"
Done.

---

## Author Comment (AC2) · 21 May 2020

**Answers to Review # 2 by Amélie Bouchat**

We would like to thank Ms. Bouchat for her careful reading and good suggestions that helped to improve the manuscript. In the manuscript, changes according to the comments are marked in yellow.

**General comments:**

A lot of equations derived in Section 2 have already been derived in previous publications, but references are not always provided (e.g. Eq. (12) is already in Lindsay and Stern, 2003; and Eq. (13) is also in Griebel and Dierking, 2018;). I would also like to note that Bruno Tremblay and I have recently submitted a paper (December 2019, currently under review at the Journal of Geophysical Research: Oceans) in which we also present and discuss the same equations as the Equations (19)-(24) in Section 2.5 in the present paper, but applied to the RADARSAT Geophysical Processor System Lagrangian drift data set only. Both H. Stern and J. Hutchings are aware of this work since they have been involved in discussions or review of this work. As mentioned to me already by H. Stern, we should cite each other's work in our revisions (I will send you a copy of the pre-print once it is accepted).

We have added a reference to Griebel and Dierking (2018) just before equation (13). We have cited Bouchat and Tremblay (2020) in Section 1, and in the first paragraph of Section 2.5, and in the first bullet of the Conclusions.

Different cases and examples for specific observation products are also discussed in the manuscript, which is useful but it is hard to extract the conclusions/main points from these examples in the text. I would therefore suggest the addition of tables/figures to convey these conclusions more clearly. For example, a table presenting the geolocation, tracking, and timing errors for the different SAR and buoy products as mentioned in Sections 3.4 and 3.5 would be a very useful reference for the reader and for future studies. Below, I also suggest presenting the dependence of the error on the number of tracked points in a graphical form so that it can be used to guide future studies on choosing how many points should be considered.

Section 3.4 was newly structured (with subsections 3.4.1 and 3.4.2) and rewritten, and a table was added showing examples for the 4 terms in equations (23) and (24), which were equations (25) and (26) in the original version of the paper.
Also section 3.5 was newly structured and rewritten to emphasize the basic question and answer to it.
Since in SAR imagery, geolocation errors must often be regarded as bias we added an explanation at the end of section 2.1 to clarify this point.

**Specific Comments:**
P1. Line 36: *"The accuracy of deformation parameters.."*
Sea-ice deformation should be defined in the introduction (i.e. shear, divergence, etc.) before mentioning their errors. It is also not clear what "deformation parameters" are. Do you refer to the shear, convergence, divergence, etc.? In this case, I would change simply to "deformations" or "deformation rates" for consistency with previous studies.
We have added two sentences at the beginning of the Introduction to explain deformation. We have edited the last paragraph of the Introduction to clarify that "deformation parameters" refers to divergence, vorticity, shear, and total deformation.

P.1 Lines 35-37: "*For buoys, errors in drift measurements depend on [...] but also by the size and shape of buoy arrays.*"
References should be added here, e.g. Hutchings et al. (2012), Griebel and Dierking (2018).
Done.

P1. Line 41: "*The issue of error estimation was repeatedly addressed in the past, scattered in a number of publications [...]*"
Also add Bouchat and Tremblay (2020, under review).
We added the phrase "and is also addressed in a more recent analysis by Bouchat and Tremblay (2020)" to the end of the sentence in question (line 46-47).

P.2 Line 50: "*for calculating errors of drift and deformation parameters, supplemented with the derivation of general-case uncertainties of divergence, vorticity, shear, and total deformation.*" Again, it is not clear what is the difference between "deformation parameters" and "divergence, vorticity, shear, and total deformation".
We have edited the sentence in question to clarify that "deformation parameters" refers to divergence, vorticity, shear, and total deformation. See the last paragraph of the Introduction.

P.4 Lines 139-142: "*If, e.g., the distance between two moving objects is closer than this, the position errors cancel and $\sigma_d^2 = \sigma_{tr}^2$ for the retrieval from a SAR image pair and $\sigma_{coord}^2 = 0$ between two buoys. Hence within a circle of 10km or less in diameter, deformation can be estimated with sufficient accuracy even if geolocation errors are high.*"
I don't understand how the geolocation errors cancel given that they are squared and add up when using the propagation of error on $d$ to obtain $\sigma_d^2 = 2\sigma_{coord}^2$. Can you explain?
Here is the mathematical explanation
$x_1 = x_{1,true} + e_{1,geo}$
$x_2 = x_{2,true} + e_{2,geo}$
$\Delta x = x_2 - x_1$
If the errors are equal, $e_{2,geo} = e_{1,geo}$, then $\Delta x = x_{2,true} - x_{1,true}$ with no error, i.e. var($\Delta x$) = 0.
If a tracking error $e_{2,tr}$ is included in $x_2$ then var($\Delta x$) = $\sigma_{tr}^2$.
In the manuscript, this question is actually fully answered in section 2.1. In the text we now refer to this section (see last paragraph of Section 2.1).

P4. Line 157: "*considering that U=...*"
I think you mean $U = d/\Delta T$ ? Otherwise, you would have to use Eq. (7) to get $\sigma_U$.
Correct. We added U = d/$\Delta$T to clarify.

P.6 Line 225: "*Throughout this section we assume that $\sigma_U = \sigma_u = \sigma_v$.*" because you assume $\sigma_{\Delta T} = 0$? If so, please mention it.
Changed to "Throughout this section we assume that $\sigma_U = \sigma_u = \sigma_v$ and $\sigma_{\Delta T} = 0$"

P.7 Eq. (18): The term $\sigma_{coord}$ should not appear in this equation since it was assumed that it is equal to zero since the beginning of this section. If not zero, then other terms should appear in Eqs. (13) & (14) and Eq. (18) to account for the error on the area and position in the strain rate definition explicitly.
You are right! We changed equation 18 as suggested.

P.8 Equations (23) and (24): These are the same as Eq. (15) and (16) presented earlier. Remove and refer to Eq. (15)-(16) instead?

Yes, the equations are formally the same. We were not sure whether it is convenient to show them in both Sects. 2.4 and 2.5 to have a full set of required equations but now removed them.

P.8 Line 295: "*Hence, the uncertainties of divergence, vorticity, shear, and total deformation differ from one another.*"
Unless $\sigma_{ux} \sim \sigma_{uy} \sim \sigma_{vx} \sim \sigma_{vy}$, then they are equal.

True, we mention this now

P.8 Line 305: The section is titled "*Typical uncertainties of deformation parameters*" but the section does not describe uncertainties but rather a short review of observed deformation rates from previous studies. If the purpose of this section is to describe observations of shear, divergence, etc. then it should be retitled, and it should also discuss the fact that the observed deformation magnitudes are closely tied with the scale of observation given that the mean deformation rate is known to decay following a power-law with increasing spatial and temporal scale (e.g. Marsan et al., 2004; Rampal et al., 2008; Stern and Lindsay, 2009; Bouchat and Tremblay, 2020 - see at the end of this document for the references if not already in your list).

The title was indeed wrong, what we meant was "Typical magnitudes of deformation parameters". The dependence of deformation magnitude on measurement scale is now explicitly mentioned with reference to Marsan. We also recognized that we did not provide the observation scale for all examples, which are now added.

P.9 Line 322: "*The first term in Eqs. (21) and (22) is smallest if, for a given area, $\sigma_A$ is at a minimum.*"
and for a given value of $u_x^2 + v_y^2$, or if $u_x^2 + v_y^2$ is also at a minimum.

We changed to "...for given area and velocity gradients…"

P.9 Section 3.2: It would be interesting to show a graph of the ratio $\sigma_A^2 / \sigma_{coord}^2$ as a function of the number of tracked points for fixed values of *A (e.g. 1 km$^2$, 10$^2$ km$^2$, 20$^2$ km$^2$, 100$^2$ km$^2$, etc.)* to complement the discussion. It seems like going from three points to four points (i.e. from triangle to square) increases the error contribution of this term, but then going from 4 points to 6 points (i.e. square to hexagon) reduces the contribution of this term to the global error. It could be useful to see this in a functional form to guide choosing (if possible) a reasonable number of tracked points to reduce the area error. This could be added in Section 3.6.

We have rearranged the order of the sentences so the progression is from triangles to squares to hexagons, and we have added a sentence at the end of the paragraph to clarify how the ratio $\sigma_A^2 / \sigma_{coord}^2$ changes with that progression. It is always proportional to area A. We think that with the new last sentence, an additional graph is not needed.

P.9 Line 340: "*For a given position error,...*" and a given tracking error
Yes. Corrected.

P.9 Line 341: "*The third term is solely dependent on the coordinate uncertainty $\sigma_{coord}$.*" No, the third (last) term also decreases with increasing area *A*.

Changed to: "The third term involving the coordinate uncertainty $\sigma_{coord}$ also decreases with increasing area A." See Section 3.3, end of first paragraph.

P.10 Line 361: "*When ice drift is retrieved from SAR images, the contribution of those terms that depend on $\sigma_{coord}/ L$ can usually be neglected.*"
Lindsay and Stern (2003) report that previous estimates of the geolocation error for the RADARSAT ScanSAR images are of the order of ~200 m, hence non-negligible when compared to the tracking error (~100 m). So it is not negligible for all SAR products. It would be worth including those estimates for RADARSAT as well since this product is often used to obtain the observed sea-ice deformation fields. In fact, in Bouchat and Tremblay (2020, under review), we show that when using all the other terms except the tracking error for the RGPS data set, the resulting error on the total deformation rates can be ~1.5 times larger than Eq. (17) in Lindsay and Stern, 2003 (or the equivalent tracking error term in your Eq. 25 and 26). In this case, the terms in $\sigma_{coord}/ L$ cannot be neglected.

Equation (30) in our paper has been re-numbered as (28a), in Section 3.6. You are correct that Lindsay and Stern (2003, hereafter LS2003) use the tracking error variance in their equation (15) (their $\varepsilon^2_i = \sigma^2_{tr}$) and their equation (16) (their $\varepsilon^2_f = \sigma^2_{tr}$). You are also correct that the last term of our equation (25) (now re-numbered as 23), namely $2\sigma^2_{tr}/(\Delta T^2 L^2)$, is the same as equation (17) in LS2003 when $L^2 = A$ and n = 4.
The geolocation uncertainty for the RGPS image is indeed about 200 m as noted in LS2003. We now include this case in our discussions in sections 3.4.1 and 3.4.2. See also our answer to P. 13 Line 500 below.

P.11 Line 395: "*with $\sigma_{tr}^2/ 2\sigma_{coord}^2$ approximately equal to $100^2/ (2\times5^2) = 200$*"
Here it is assumed that $\sigma_{coord} = 5$ m for RADARSAT ScanSAR images, but Lindsay and Stern (2003) mention a geolocation error that is on the order of ~ 200 m (see also comment above). Can you indicate where the value of $\sigma_{coord} = 5$ m was taken from?

This expression has now been deleted. See Section 3.4.2, third paragraph.

P.9-13 Sections 3.4 and 3.5: These two sections focus on estimation of errors for the divergence and vorticity, and compare the contribution of different terms to the total error on divergence and vorticity. However, shear and total deformation rates are often larger than divergence (see increased probability of larger deformation rates in PDFs of shear vs divergence in e.g. Bouchat and Tremblay 2017, or Stern et al. 1995). How does the interpretation of the importance of each term in the error formulation change when considering the error on shear and total deformation instead of divergence and vorticity? For this, it could also be useful to present the expanded version of Eq. (23) and (24).

We decided to add alternative expressions for equations (15) and (16): The principal direction of shear is given by the angle $\phi = \frac{1}{2}$ arctan $((u_y + v_x) / (u_x - v_y))$. Therefore, $\sigma^2_{shr} = \cos^2(2\phi) \sigma^2_{div} + \sin^2(2\phi) \sigma^2_{vrt}$, which is now our equation (15b). The error variance of shear is a weighted average of the error variances of divergence and vorticity, which is now mentioned in the conclusions. (The weights add up to 1: $\cos^2 + \sin^2 = 1$). Total deformation: $\varepsilon_{tot} = $ sqrt $(\varepsilon^2_{div} + \varepsilon^2_{shr})$. Think of divergence and shear as rectangular coordinates x and y. Then in polar coordinates, the radial distance or magnitude is $\varepsilon_{tot}$, and the angular

coordinate is given by $\theta = \arctan(\varepsilon_{shr} / \varepsilon_{div})$.  With this definition, our equation (16a) becomes equation (16b): $\sigma^2_{tot} = \sin^2(\theta) \sigma^2_{shr} + \cos^2(\theta) \sigma^2_{div}$. The error variance of total deformation is a weighted average of the error variances of shear and divergence.  (The weights add up to 1 again).
We think this clarifies the interpretation of the importance of each term in the error formulation of shear and total deformation.

I also found it hard/confusing to follow all the examples presented at the end of section 3.5. A lot of different cases and numbers are presented and it is easy to get lost in the conclusions that should be retained. A visual aid (such as a table or graph) that gathers the essential points that are supposed to be conveyed by these examples could be added for more clarity.
We changed section 3.5 by dividing it into subsections, emphasizing the essential points through the title and main conclusions at the end.

P.13 Line 489: "*Let L' be the length of each side of the big square (Fig 5).*" L' is not defined on Figure 5. And "*big square*" = "window" ?
We changed Fig. 5 and the text to clarify.

P.13 Line 490: "*Because of the enclosed grid cells we can divide each side of the square window into N segments of equal length.*"
Is Eq. (30) derived here only valid when the grid cells are not moving and of equal length? For a Lagrangian grid, the cells are not necessarily of equal size and therefore Eq. (30) would not apply, and one would still need to evaluate the full expression in (12), correct?
Yes! (Now equations 28)

P.13 Line 500: "*we can rewrite Eq. (30) as $\sigma_A^2 = \sigma_{coord}^2(n-2)L^2$ which is Eq. (16) in Lindsay and Stern (2003)*"
Equation (16) in Lindsay and Stern (2003) uses the tracking error $\sigma_{tr}^2$ instead of the geolocation error as considered here $\sigma_{coord}^2$ . It is confusing because in Section 3.4, it is mentioned (line 360) that the last term of Eq. (25) term depending on the tracking error is the same as Eq (17) in Lindsay and Stern (2003), but Eq. (17) in Lindsay and Stern is derived using their Eq. (16) which is now assumed to be using the geolocation error here... Can you please clarify? Also, please indicate if there is a mistake in Lindsay and Stern (2003).
The calculation of sea-ice deformation in SAR images is based on two images separated in time – call them image #1 and image #2.  When we refer to coordinates (x, y) in these images, it's important to distinguish image #1 from image #2.  Let $(x_i, y_i)$ refer to a point in image #1, and let $(x'_i, y'_i)$ refer to the corresponding point in image #2, as determined by a tracking algorithm.  So image #1 has unprimed coordinates, and image #2 has primed (') coordinates.

Let's start at the most basic level: errors in position.  Following Holt et al. (1992), the measured x-coordinate in image #1 has a geolocation error:
$x = x_{true} + e_{geo}$
and the measured x-coordinate in image #2 has a geolocation error and a tracking error:
$x' = x'_{true} + e'_{geo} + e'_{tr}$

Assume zero-mean uncorrelated errors, and equal geolocation error variances in the two images. Then: $\sigma^2_x = \sigma^2_{geo}$ and $\sigma^2_{x'} = \sigma^2_{geo} + \sigma^2_{tr}$.

Look at the area formula, equation (9) in our paper:

$A = \frac{1}{2} \Sigma (x_i y_{i+1} - y_i x_{i+1})$

With unprimed coordinates, this is the area in image #1. The development of the error in A proceeds according to equations (10), (11), and (12). Going from equation (11) to (12), $\sigma^2_x = \sigma^2_y = \sigma^2_{coord}$ is the geolocation error variance.

Suppose we write down the area of the region in image #2:

$A' = \frac{1}{2} \Sigma (x'_i y'_{i+1} - y'_i x'_{i+1})$

Now proceed through equations (10), (11), and (12) with primed (') coordinates for image #2. Going from equation (11) to (12), $\sigma^2_{x'} = \sigma^2_{y'} = \sigma^2_{coord} + \sigma^2_{tr}$ which is the geolocation error variance plus the tracking error variance.
So we could write:

$\sigma^2_A = (\sigma^2_{coord} / 4) \Sigma [(x_{i+1} - x_{i-1})^2 + (y_{i+1} - y_{i-1})^2]$      (image #1)     (12a)

$\sigma^2_{A'} = ((\sigma^2_{coord} + \sigma^2_{tr}) / 4) \Sigma [(x'_{i+1} - x'_{i-1})^2 + (y'_{i+1} - y'_{i-1})^2]$    (image #2)    (12b)

These equations also apply to an array of buoys, but of course in that case $\sigma^2_{tr} = 0$.

For SAR images, with $\sigma^2_{coord} = 0$, we have:
$\sigma^2_A = 0$                                         (image #1)
$\sigma^2_{A'} = (\sigma^2_{tr} / 4) \Sigma [(x'_{i+1} - x'_{i-1})^2 + (y'_{i+1} - y'_{i-1})^2]$      (image #2)    (*)

For a buoy array, with $\sigma^2_{tr} = 0$, we have:
$\sigma^2_A = (\sigma^2_{coord} / 4) \Sigma [(x_{i+1} - x_{i-1})^2 + (y_{i+1} - y_{i-1})^2]$      (at time t)
$\sigma^2_{A'} = (\sigma^2_{coord} / 4) \Sigma [(x'_{i+1} - x'_{i-1})^2 + (y'_{i+1} - y'_{i-1})^2]$      (at time t + ΔT)

Equation (15) in Lindsay and Stern (2003) is based on equation (*) above for image #2. That's why the tracking error variance (only) appears in their equation (16).
Our equation (12) is based on SAR image #1, or a buoy array, and therefore contains the geolocation error variance (only), not the tracking error variance.
Note that our equation (12) says $\sigma^2_A = 0$ for SAR images because $\sigma^2_{coord} = 0$.
We added a short explanation of this issue below Eq. 28a.

P.13 Line 506: *"Note that $\sigma_{coord}^2$ for the right triangle is $\sigma_{coord}^2 L'^2$ for N = 1, $1.25\sigma_{coord}^2 L'^2$ for N = 2, $\sigma_{coord}^2 L'^2$ for N = 3, and $0.8125\sigma_{coord}^2 L'^2$ for N = 4, i.e. for N = 2 the uncertainty increases, N = 3 and N = 1 reveal the same uncertainty, and first with N = 4, the uncertainty can be reduced."*
It could be worth presenting this discussion using a graphical form, for more clarity. See also previous comment for P.9 Section 3.2.

We have rewritten the sentence to improve clarity. It now follows equation (28b).

P.13 Line 513: *"Hence the uncertainty of the area increases when elementary cells are combined. However, since also the cell area increases by a factor of $N^2$, the single terms in Eqs. (13) – (22) that include the factor $A^{-2}$ decrease."*
Which one wins? Is it better in the end to aggregate cells?
Good point! This was indeed explained only vaguely. We rephrased the end of this paragraph and pointed out the decrease of the uncertainty of deformation parameters when using combinations of elementary cells versus the need for recognizing local variations of deformation by using the elementary cells. See Section 3.6.

P.14 Line 517: *"For buoy arrays it may be of advantage to use a larger number of buoys along the outline of a polygon."*
Couldn't SAR drift fields also be derived using triangle cells (in principle) and the discussion regarding Eq. (31) could therefore apply to both SAR and buoy applications?
Yes, we mention this now. See Section 3.6.

P.14 Line 535: "*To calculate the number of chords that is required to fulfill Eq. (32), we demand that $n'_{sc}$ (1+e) = 2πr, with n'= n/2, and e the error.*"
This is unclear; "e" is the error on/of what?
Now explained in the text: "e is the error between the perimeter of a regular polygon and a circle." See Section 3.6, last paragraph.

P.14 Eq (33): It is not clear to me how you obtain this result. I can see that it probably involves a Taylor series expansion of u(x,y) however if I do this expansion around $(x_k, y_k)$, e.g.:

$$u(x,y) = u(x_k,y_k)+(x-x_k)u_x+(y-y_k)u_y+\frac{1}{2}\left[(x-x_k)^2u_{xx}+2(x-x_k)(y-y_k)u_{xy}+(y-y_k)^2u_{yy}\right]+[...]$$

where the derivatives are evaluated $(x_k, y_k)$. Then I evaluate u(x,y) at $(x_{k+1}, y_{k+1})$ and use the same definitions of $\Delta x_k$ and $\Delta y_k$ as in the manuscript, and I get:

$$u_{k+1} = u_k+\Delta x_k u_x+\Delta y_k u_y+\frac{1}{2}\left[\Delta x_k^2 u_{xx}+2\Delta x_k\Delta y_k u_{xy}+\Delta y_k^2 u_{yy}\right]+[...]$$

Such that, I get:

$$\frac{1}{2}\left(u_{k+1}+u_k\right)\Delta y_k = u_k\Delta y_k+\frac{1}{2}\Delta x_k\Delta y_k u_x+\frac{1}{2}\Delta y_k^2 u_y+\frac{\Delta y_k}{4}\left[\Delta x_k^2 u_{xx}+2\Delta x_k\Delta y_k u_{xy}+\Delta y_k^2 u_{yy}\right]+[...]$$

So I see that my last term here is similar to your definition of $e_k$ but I don't know how to get there. Can you clarify?

Yes, we can clarify. (Please note that the equation is now numbered (31), not (33)). The trapezoid rule of integration is:

$$\int_a^b f(t)dt = (b-a)\left[\frac{f(a)+f(b)}{2}\right] + E$$

where the first term on the right-hand side is the estimate of the integral, and the second term $E$ is the error. The error is the difference between the true integral (left-hand side) and the estimate. The error is given by:

$$E = -\frac{(b-a)^3}{12}f''(\xi)$$

where $f''(\xi)$ is the second derivative of $f(t)$ evaluated at some point $\xi$ on the interval $(a,b)$. You can find this result in any book on numerical analysis, or Wikipedia
https://en.wikipedia.org/wiki/Trapezoidal_rule

Now we apply these formulas to the problem at hand. We want to evaluate the segment of the contour integral $\oint u\,dy$ that goes from $(x_k,y_k)$ to $(x_{k+1},y_{k+1})$. We parameterize that segment as:

$x(t) = x_k + (x_{k+1}-x_k)\,t$

$y(t) = y_k + (y_{k+1}-y_k)\,t$

where the parameter t runs from 0 to 1. We also make the definitions:

$\Delta x_k = x_{k+1}-x_k$

$\Delta y_k = y_{k+1}-y_k$

and just for convenience let's drop the subscript $k$ and write these as $\Delta x$ and $\Delta y$. So we have $dx/dt = \Delta x$ and $dy/dt = \Delta y$.

Now the $k^{th}$ segment of the contour integral is:

$$\int_0^1 u(x(t),y(t))\frac{dy}{dt}dt = \int_0^1 u(x(t),y(t))\Delta y\,dt$$

Referring back to the trapezoid rule, we have $f(t) = u(x(t),y(t))\Delta y$, and $a = 0$ and $b = 1$. The estimate of the integral is therefore $\frac{1}{2}\,(f(0) + f(1)) = \frac{1}{2}\,(u(x_k,y_k) + u(x_{k+1},y_{k+1}))\Delta y$, as we wrote just before equation (31). The error term is $E = -(1/12)\,f''(\xi)$. Taking derivatives of $f(t)$, we have:

$f'(t) = ((\partial u/\partial x)(dx/dt) + (\partial u/\partial y)(dy/dt))\Delta y = (u_x\Delta x + u_y\Delta y)\Delta y$

$f''(t) = (u_{xx}\Delta x^2 + 2u_{xy}\Delta x\Delta y + u_{yy}\Delta y^2)\Delta y$

and equation (31) follows from this. We added a reference to the trapezoid rule and its error term (Atkinson, 1989) following equation (31). We think it is not necessary to include the above mathematical detail.

P.16 Line 591: "*(see above)*"
Not clear to what this is referring to.

We deleted "(see above)". It is easy enough for the reader to refer back to equation (5).

P.16 Line 624: "*For a general configuration of points, the three methods give different estimates.* "
Have you obtained numerical estimates for examples using each method? How much do they differ?
They differ slightly. The reason for this discussion here is only to make the reader aware that more methods exist for calculating deformation.

P.17 Section 4 Conclusions:
The first point has also been shown in Bouchat and Tremblay (2020, under review). The second point should also mention the exception for RGPS.
We added "These results agree with the recent work of Bouchat and Tremblay (2020)." We also rephrased bullet 2.

P. 19 Line 725: "*RADARSAR*" should be RADARSAT
Corrected

**Formatting and writing suggestions:**

P.1 Line 16: "*in an array.*" → "in an array of buoys."
We are speaking very generally here of "position sensors in an array"

P.1 Line 19: "*also a tracking error has to be considered.*" → "a tracking error also has to be considered."
Done

P.1, Line 24: "*the magnitudes of deformation parameters*" → "the magnitude of deformation parameters."?
Done

P1. Line 33: "*sea ice mapping*" → "sea-ice mapping".
Done

P. 2 Line 46: "*truncation error*" → change to "boundary-definition errors", or add it in parentheses to link with previously-used formulation? (or indicate why the previous formulation is incorrect).
Done

P.4 Line 134: "*one needs to consider position and tracking uncertainties $\sigma_{coord}^2$ and $\sigma_{tr}^2$.*" →

"one needs to consider position and tracking uncertainties, *i.e.* $\sigma_{coord}^2$ and $\sigma_{tr}^2$ *respectively.* "

Done

P.6 Line 208: "*The cell covers m x m square-shaped pixels.*" → "The cell covers m x m square-shaped pixels of resolution $\Delta x$ ."
We used the phrase: "of side length $\Delta x$".

P.6 Lines 236-237: "*the sum of variances of the left term*" → "the sum of variances of the *first* term"? And "*the sum in the right term*" → "the sum in the *second* term"?
This sentence does not exist anymore

P.8 Line 286: "*For the shear one obtains...*" → "For the shear, one obtains..." P8. Eq. (23): x and y in $\sigma_{u_x}$, $\sigma_{u_y}$, $\sigma_{v_x}$, and $\sigma_{v_y}$ should be subscripts.
Sentence does not exist anymore. Subscripts of subscripts are too small.

P.9 Line 343: "*In the following discussion we assume that position data of all buoys are exactly synchronized but also discuss an example for which this was not the case.*"
Add reference to section 3.5 at the end of this sentence?
Done

P.13 Line 485: "*the uncertainties have to be calculated numerically.*" → "the uncertainties have to be calculated numerically using Eq. (12)"?
Good hint. Done

P.14 Line 515: "*can be considered*" → "can *also* be considered"?
Since we changed the text here, this does not fit.

**Figures:**
Fig. (2): The figure is blurry.
Yes, the figure was blurry in the posted pdf version of the paper, but not in the original Word doc. Will be corrected in the final print.

Fig. (5): Please indicate in the label what are the blue and green lines.
Done

---

## Author Comment (AC3) · 21 May 2020

[revised manuscript text omitted]
^2)}{b^2a^2} (u_x^2 + v_y^2) + \frac{(2\sigma_{tr}^2 + \sigma_{tr}^2)(a^2+b^2+c^2)}{b^2a^2} (u_x^2 + v_t^2) + \frac{(2\sigma_{tr}^2 + \sigma_{tr}^2)(a^2+b^2+c^2)}{b^2a^2} (u_x^2 + v_t^2) + \frac{(2\sigma_{tr}^2 + \sigma_{tr}^2)(a^2+b^2+c^2)}{b^2a^2} (u_x^2 + v_t^2) + \frac{(2\sigma_{tr}^2 + \sigma_{tr}^2)(u_x^2 + v_t^2)}{b^2a^2} (u_x^2 + v_t^2) + \frac{(2\sigma_{tr}^2 + \sigma_{tr}^2)(u_x^2 + v_t^2)}{b^2a^2} (u_x^2 + v_t^2) + \frac{(2\sigma_{tr}^2 + \sigma_{tr}^2)(u_x^2 + v_t^2)}{b^2a^2} (u_x^2 + v_t^2) + \frac{(2\sigma_{tr}^2 + \sigma_{tr}^2)(u_x^2 + v_t^2)}{b^2a^2} (u_x^2 + v_t^2) + \frac{(2\sigma_{tr}^2 + v_t^2)(u_x^2 + v_t^2)}{b^2a^2}$$

[revised manuscript text omitted]